# Effective Instruction Parsing Plugin for Complex Logical Query Answering on Knowledge Graphs

Anonymous Authors (Paper ID: 67)

## Abstract

Knowledge Graph Query Embedding (KGQE) aims to embed First-Order Logic (FOL) queries in a low-dimensional KG space for complex reasoning over incomplete KGs. To enhance the generalization of KGQE models, recent studies integrate various external information (such as entity types and relation context) to better capture the logical semantics of FOL queries. The whole process is commonly referred to as *Query Pattern Learning* (QPL). However, current QPL methods typically suffer from the pattern-entity alignment bias problem, leading to the learned defective query patterns limiting KGQE models' performance. To address this problem, we propose an effective Query Instruction Parsing Plugin (QIPP) that leverages the context awareness of Pre-trained Language Models (PLMs) to capture latent query patterns from code-like query instructions. Unlike the external information introduced by previous QPL methods, we first propose code-like instructions to express FOL queries in an alternative format. This format utilizes textual variables and nested tuples to convey the logical semantics within FOL queries, serving as raw materials for a PLM-based instruction encoder to obtain complete query patterns. Building on this, we design a query-guided instruction decoder to adapt query patterns to KGQE models. To further enhance QIPP's effectiveness across various KGQE models, we propose a query pattern injection mechanism based on compressed optimization boundaries and an adaptive normalization component, allowing KGQE models to utilize query patterns more efficiently. Extensive experiments demonstrate that our plug-and-play method[1] improves the performance of eight basic KGQE models and outperforms two state-of-the-art QPL methods.

## CCS Concepts

• **Computing methodologies → Knowledge representation and reasoning**; • **Information systems → Data mining**.

## Keywords

Knowledge Graph, Complex Query Answering, Pre-trained Language Model, Query Pattern Learning

## 1 Introduction

Knowledge Graph (KG) question answering [14, 29, 43] aims to reason on KGs for answering a given question. However, the inherent ambiguity in natural language (NL) questions often obstructs their ability to accurately convey reasoning logic. Thus, previous methods [1] usually transform NL questions into First-Order Logic (FOL) queries for explicit reasoning on KGs. As shown in Figure 1(b), the FOL query can intuitively express the reasoning logic of the corresponding NL question (Figure 1(a)) through various logical operations, such as existence quantification (∃), conjunction (∧), disjunction (∨), and negation (¬).

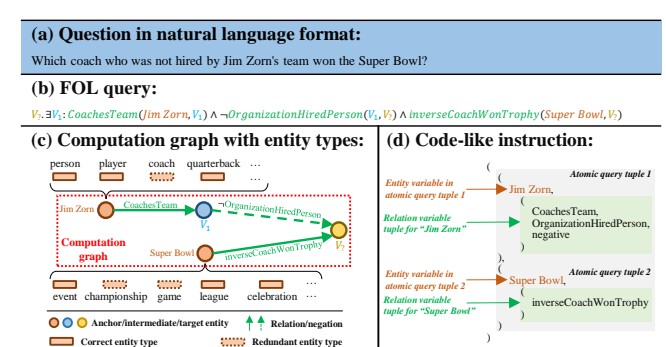

**Figure 1: Four forms of a complex question. (a) is an NL question and (b) is the FOL query of (a). (c) and (d) are the computation graph and the code-like instruction of (b), respectively.**

To better capture the logical semantics in FOL queries, Knowledge Graph Query Embedding (KGQE) [20, 28, 30] is proposed. Specifically, KGQE models extract a computation graph (the red dashed box in Figure 1(c)) of an FOL query from a KG and embed this query into a low-dimensional KG space according to the entity-relation order in the computation graph. Through the computation graph, KGQE models can identify entities close to a given query as answers.

In general, computation graphs can convey complex logical patterns of FOL queries, such as the projection and intersection patterns in Figure 2. These patterns are beneficial for a KGQE model to gain a deeper understanding of the logical semantics of FOL queries. However, due to the incompleteness of KGs in reality, an extracted computation graph is difficult to fully display the logical pattern of a query. Therefore, recent studies propose *Query Pattern Learning* (QPL) to solve this problem. QPL incorporates external information (such as entity types [19] and relation context [22]) into a computation graph to make up for the missing logical patterns. **Our study focuses on proposing a novel QPL plugin to improve the pattern semantic awareness of various KGQE models applied on incomplete web KGs, which is highly relevant to the track "Semantics and Knowledge"**.

Although current QPL plugins [19, 22] can enhance KGQE models to some extent, they still suffer from the *pattern-entity alignment bias* problem. As demonstrated by *Fu et al.* [15], assigning randomly sampled types to anchor entities in an FOL query can introduce redundant types (as shown in Figure 1(c)). Consequently, QPL plugins may derive noisy query patterns from these redundant types. Moreover, since the semantics of an entity are more concrete than those of a query pattern, query answering is more sensitive to redundant types than query pattern recognition. This sensitivity can mislead KGQE models during query answering, which is the primary manifestation of the pattern-entity alignment bias problem. Therefore, effectively converting the FOL query into a format more suitable for QPL is the key to addressing the problem.

---
[1]Our anonymous codes are available at https://anonymous.4open.science/r/QIPP-CF71

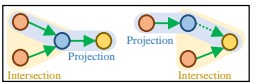

**Projection pattern**   **Intersection pattern**   **Projection + intersection pattern**

🔴 Anchor entity   🔵 Intermediate entity   🟡 Target entity (answer)   ⬆⬆ Relation/negation

**Figure 2: Several query patterns illustrated by computation graphs. The projection pattern indicates that an FOL query is a single/multi-hop chain query. The intersection pattern represents that an FOL query is the intersection of multiple chain query.**

Inspired by the achievements of Pre-trained Language Models (PLMs) on instruction graph reasoning [37], we propose to represent an FOL query in a textual code-like instruction format (as shown in Figure 1(d)), which enables a PLM to learn relevant query patterns for query answering. This approach is effective for several reasons: (i) variable names in the instructions precisely represent the semantics of entities and relations without the need for additional types or contexts for information expansion; (ii) compared to FOL queries that contain unintelligible logical symbols, nested tuples in code-like instructions offer a clearer representation of entity-relation projections and the conjunction logic between atomic queries; and (iii) by leveraging the contextual understanding of PLMs, the variable descriptions and nested tuples in instructions can be more effectively utilized for QPL.

Based on the above analysis, we propose a plug-and-play Query Instruction Parsing Plugin (QIPP) for KGQE models. In specific, the proposed QIPP first employs a PLM-based instruction encoder to capture entity/relation descriptions and logic structures from code-like query instructions. This process establishes a relatively complete information set for query pattern learning. Next, a query-guided instruction decoder is designed to allow the query embedding obtained from a KGQE model to parse the desired query patterns from the information set. Furthermore, to adapt the parsed query patterns to the optimized parameter boundaries of different KGQE models, we propose a query pattern injection mechanism based on compressive optimization boundaries and an adaptive normalization component, which further enhances the role of QIPP across various KGQE models.

Our main contributions can be summarized as follows:

- We propose an effective plug-and-play Query Instruction Parsing Plugin (QIPP) for KGQE. QIPP captures the intrinsic logical principles of FOL queries through a PLM-based encoder and a query-guided decoder, which eliminates the pattern-entity alignment bias caused by the existing QPL plugins.
- We design code-like instructions, an alternative format for FOL queries, which can provide a complete query pattern learning environment for QIPP without introducing external noise.
- To enhance the adaptability of the proposed QIPP across different KGQE models, we propose a query pattern injection mechanism based on compressive optimization boundaries and an adaptive normalization component;
- Extensive experiments on three widely used benchmarks demonstrate that our proposed QIPP significantly improves the performance of eight basic KGQE models and outperforms two state-of-the-art QPL plugins.

## 2 Preliminaries

Before elaborating on our proposed method, let's first introduce the task background of KGQE models and our main objective.

### 2.1 Knowledge Graph and FOL Query

Let $\mathcal{G} = (\mathcal{E}, \mathcal{R})$ be a KG consisting of an entity set $\mathcal{E} = \{e_n\}_{n=1}^{N}$ and a directed relation set $\mathcal{R} = \{r_m\}_{m=1}^{M}$. In $\mathcal{G}$, a relation assertion $r_m(\mathcal{E}_h, \mathcal{E}_t)$ represents that any entity in a head entity set $\mathcal{E}_h \subseteq \mathcal{E}$ can be mapped to any entity in a tail entity set $\mathcal{E}_t \subseteq \mathcal{E}$ through a relation $r_m \in \mathcal{R}$.

According to the definition of [32], an FOL query can be defined by the following disjunctive paradigm (*i.e.*, the disjunction of conjunction atomic queries):

$$q[\mathcal{E}_?] = \mathcal{E}_?.\exists \mathcal{E}_1, \mathcal{E}_2, ..., \mathcal{E}_k : \vee_{i=1}^{I}(\wedge_{j=1}^{J} q_{ij}),$$

$$q_{ij} = \begin{cases} r(\mathcal{E}_a, \mathcal{E}^*) \ \ \text{or} \ \ \neg r(\mathcal{E}_a, \mathcal{E}^*) \\ r(\mathcal{E}', \mathcal{E}^*) \ \ \text{or} \ \ \neg r(\mathcal{E}', \mathcal{E}^*) \end{cases},$$

where $\mathcal{E}^* \in \{\mathcal{E}_?, \mathcal{E}_1, ..., \mathcal{E}_k\}$, $\mathcal{E}' \in \{\mathcal{E}_1, ..., \mathcal{E}_k\}$, $r \in \mathcal{R}$, $\mathcal{E}^* \neq \mathcal{E}'$, $\mathcal{E}_1, ..., \mathcal{E}_k \subseteq \mathcal{E}$ and $\mathcal{E}' \subseteq \mathcal{E}$ are the sets of existentially quantified bound variables (*i.e.*, intermediate entities) and the set of target entities, respectively. An indivisible atomic query $q_{ij}$ is represented by a relation assertion itself or a negation operation of a relation assertion. $\mathcal{E}_a \subseteq \mathcal{E}$ is a set of anchor entities that appear in $q[\mathcal{E}_?]$. Given a specific FOL query $q[\mathcal{E}_?]$, our purpose is to find out a $\mathcal{E}_?$ that satisfies the condition of $q[\mathcal{E}_?]$ as true.

### 2.2 Knowledge Graph Query Embedding

KGQE aims to embed an FOL query in a low-dimensional KG space according to the query's computation graph. Let $q$, $g$, and $f_{e_n}$ be an FOL query $q$, the computation graph $g$ of $q$, and the embedding of $e_n \in \mathcal{E}$, respectively, we first use a KGQE model $QE(\cdot)$ to obtain the query's embedding $f_q$ based on $g$. Next, a score function $\mathcal{S}(\cdot)$ is used to calculate the embedding similarity between $q$ and each $e_n \in \mathcal{E}$. Then, a KGQE model defines a specific reasoning operation $opt(\cdot)$ to predict the answer $e_i \in \mathcal{E}$ of $q$. The overall framework of a KGQE model can be summarized as:

$$e_i \leftarrow \underset{f_q=QE(q|g)}{opt} (e_i \in \mathcal{E} | \{\mathcal{S}(f_q, f_{e_n}) | e_n \in \mathcal{E}\}_{n=1}^{N}).$$

According to the reasoning mechanism of $opt(\cdot)$, existing KGQE models can be divided into two categories: *end-to-end KGQE models* and *iterative KGQE models*.

**End-to-end KGQE models** directly obtain the embedding of an FOL query $q$. At this point, $opt(\cdot)$ is represented by $\max(\cdot)$, which means the entity $e_i$ most similar to $q$ is used as the predicted answer.

**Iterative KGQE models** first pretrain a KG Embedding (KGE) model (such as ComplEx [34]) as $QE(\cdot)$ for atomic query embedding, and then split an FOL query into multiple atomic queries that are sequentially input into $QE(\cdot)$. At this moment, $opt(\cdot)$ relies on a specific T-norm inference framework [17, 23, 24] based on beam search [5] or tree optimization [7]. In this case, $opt(\cdot)$ iteratively finds the optimal result $e_i$ of the given atomic query $q$ and regards $e_i$ as the anchor entity for the next atomic query. Once the iteration is complete, the final $e_i$ is used as the predicted answer for the complex query.

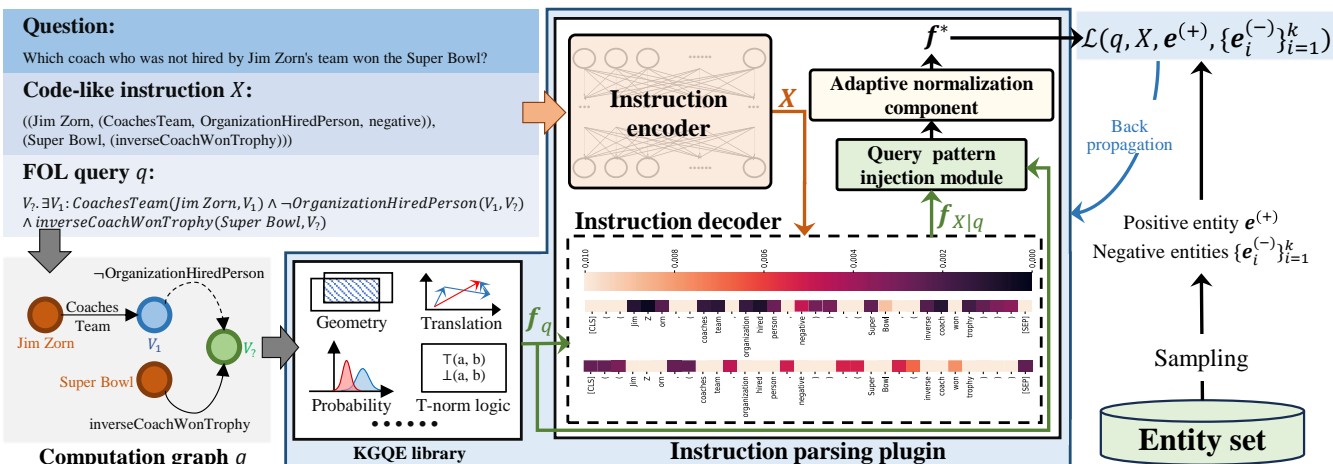

**Figure 3: The overall framework of QIPP. We first transform the given question into an FOL query $q$ and a code-like instruction $X$. A specific KGQE model in the library is used to represent $q$ as the initial query embedding $f_q$ based on the computation graph $g$. Then, we process $X$ into $X$ through a PLM-based instruction encoder. Next, $f_q$ is used to decode meaningful query pattern $f_{X|q}$ from $X$. Finally, we fuse $f_q$ and $f_{X|q}$ into query embedding $f^*$ containing pattern information through a query pattern injection module and an adaptive normalization component for subsequent model training and query answering.**

## 2.3 QPL Plugin

When $e_i$ is the ground truth of $q$, the purpose of a KGQE method is to optimize $QE(q|g)$ such that $\mathcal{S}(f_q, f_{e_i})$ approaches $\max P(e_i|q)$. However, relying solely on the corresponding $g$, existing KGQE models are difficult to predict the ground truth of $q$ on an incomplete KG. Therefore, QPL plugins [19, 22] usually introduce external information to enrich $g$, so that $f_q$ can represent more general logical semantics of $q$ (i.e., query pattern information), thereby enhancing the generalization of KGQE models. Let $X$ be the external information, the optimization of $\max P(e_i|q)$ can be transformed into:

$$\max P(e_i|q,X) \propto \underset{f_q=QE(q|g,X)}{opt} (e_i \in \mathcal{E}|\{\mathcal{S}(f_q,f_{e_n})|e_n \in \mathcal{E}\}_{n=1}^N), \quad (1)$$

Here, the current QPL plugins are mainly used for end-to-end KGQE models.

## 2.4 Our Objective

As analyzed in Section 1, existing QPL plugins are affected by pattern-entity alignment bias primarily due to their insufficient methods for representing and learning external information $X$. Our objective is to build a more effective QPL plugin to overcome this issue and ensure the proposed plugin can adapt to both the end-to-end and iterative KGQE models outlined in Section 2.2.

To achieve this objective, we can process $P(e_i|q,X)$ in Eq. (1) into $\frac{P(X|q,e_i)P(e_i|q)}{P(X|q)}$. Because $P(X|q) = \sum_{e_j \in \mathcal{E}} P(X|q,e_j)P(e_j|q)$ is a normalization term, we pay more attention to the modeling of $P(X|q,e_i)P(e_i|q)$. Therefore, our main objective is:

$$\max P(e_i|q,X) \propto \max P(X|q,e_i)P(e_i|q). \quad (2)$$

Since $P(e_i|q)$ can be directly obtained through a specific $QE(\cdot)$, we focus on analyzing the modeling method of $P(X|q,e_i)$, i.e., the instruction parsing plugin described in Section 3.

## 3 Query Instruction Parsing Plugin

As outlined in Section 2.4, the instruction parsing plugin is used to inject the pattern information of the query into its embedding to achieve more accurate target entity prediction. Figure 3 provides the framework of our proposed QIPP.

## 3.1 Code-like Instruction

As described in Section 1, compared to the FOL query format that contains unintelligible logical symbols, the code-like instructions use textual codes to more clearly express the information of a query's elements and the hierarchical structure of a complex query. Table A1 in **Appendix A1.1** presents the code-like instruction format for each query type we use.

In addition, following the operations of previous studies on disjunctive FOL queries ("2u" and "up" in Table A1) [18, 31, 32], we construct the code-like instructions only for all non-extensible conjunctive sub-queries in the disjunctive query for a KGQE model. The final prediction for the disjunctive query is obtained by taking the union of the candidate target entity sets generated for each conjunctive sub-query.

## 3.2 Instruction Encoder and Decoder

*3.2.1 **Instruction encoder**.* Inspired by the effectiveness of PLMs in understanding the structure of programming languages [3], we construct a BERT-based instruction encoder [13] shown in Eq. (3), which learns the context of instructions to build a pattern learning space for a query:

$$X = INST_\Theta(X), \quad (3)$$

where the code-like instruction $X = \{x_z\}_{z=1}^Z$ is a sequence that consists of $Z$ tokens, $INST_\Theta(\cdot)$ is a BERT-based instruction encoder with pre-trained parameters $\Theta$, and $X = \{x_z \in \mathbb{R}^{1 \times F_1}\}_{z=1}^Z$ is the

result of $INST_{\Theta}(X)$ that consists of $Z$ $F_1$-dimensional token embeddings.

Considering the differences in context learning scales across different layers of BERT [21], we aim for $INST_{\Theta}(\cdot)$ to focus more on the hierarchical structure within $X$ rather than the abstract representation of a query. Therefore, we define $\Theta$ to include only BERT's token embedding layer and shallow encoding layers. The exact number of layers will be analyzed in detail in Section 4.4.4.

*3.2.2 **Instruction decoder**.* The primary purpose of the instruction decoder is to allow $q$ to retrieve for more representative pattern information in the corresponding instruction $X$, thereby optimizing the representation of $q$. Therefore, we adopt a pattern information retrieval method based on the multi-head attention mechanism to construct an instruction decoder. Let $f_{X|q}$ be the decoded output retrieved by $q$ in $X$, $W_Q \in \mathbb{R}^{\frac{F_2}{H} \times \frac{F_2}{H}}$, $W_K, W_V \in \mathbb{R}^{\frac{F_1}{H} \times \frac{F_2}{H}}$, and $W_O \in \mathbb{R}^{F_2 \times F_2}$ are four trainable parameter matrices. When the embedding $f_q \in \mathbb{R}^{1 \times F_2}$ of $q$ is obtained from a specific KGQE model $QE(\cdot)$, $f_{X|q}$ can be represented as:

$$f_{X|q} = (\overset{H}{\underset{h=1}{\|}} \sigma(\frac{f_q^{(h)} W_Q^{(h)} (X^{(h)} W_K^{(h)})^T}{\sqrt{F_2/H}}) X^{(h)} W_V^{(h)}) W_O, \quad (4)$$

where $H$ is the number of attention heads, $\|$ is a column-wise vector concatenation operation, $\sigma(\cdot)$ is a softmax function, and $f_q^{(h)} \in \mathbb{R}^{1 \times \frac{F_2}{H}}$ and $X_q^{(h)} \in \mathbb{R}^{Z \times \frac{F_1}{H}}$ are the embedding chunks of $f_q$ and $X$ in the $h$-th attention head, respectively.

## 3.3 Query Pattern Injection

In this section, we discuss how to solve $P(X|q, e_i)P(e_i|q)$ in Eq. (2) by injecting query pattern information. According to Eq. (4), $f_{X|q}$ is the query pattern obtained from $X$ based on $q$, while $P(X|q, e_i)$ theoretically refers to the query pattern obtained from $X$ under the conditions of both $q$ and $e_i$. To measure the correlation between $f_{X|q}$ and the condition $e_i$, we approximate $P(X|q, e_i)$ to a similarity between $f_{X|q}$ and $f_{e_i}$:

$$P(X|q, e_i) = \mathcal{S}(f_{X|q}, f_{e_i}). \quad (5)$$

Similarly, according to the description in Section 2.4, $P(e_i|q)$ can be expressed as Eq. (6):

$$P(e_i|q) = \mathcal{S}(f_q, f_{e_i}). \quad (6)$$

Let $\mathcal{S}(a, b) = \exp(a \circ b)$, where $\circ$ is a vector distance function in a specific $QE(\cdot)$. Combining Eqs. (5) and (6), we can convert Eq. (2) into Eq. (7):

$$\min \mathcal{S}(f_{X|q}, f_{e_i}) \mathcal{S}(f_q, f_{e_i}) \propto \min(f_{X|q} \circ f_{e_i} + f_q \circ f_{e_i}) \quad (7)$$

To enhance the generalization of query pattern injection in the KGQE model, we construct a compressed optimization boundary for Eq. (7) based on the law of parsimony (Occam's Razor) [8, 16]. Instead of supervising $f_q$ and $f_{X|q}$ separately, we use $f_{X|q} + f_q$ as a unified supervised signal for entity prediction. This approach can be proven through the following theorem.

**Theorem 3.1.** *When $f_{X|q}$ and $f_{X|q} + f_q$ satisfy a specific constraint of $QE(\cdot)$ on the values of $f_q$ and $f_{e_i}$, the optimized parameter boundary of $(f_{X|q} + f_q) \circ f_{e_i}$ is more compact than that of $f_{X|q} \circ f_{e_i} + f_q \circ f_{e_i}$. Minimizing $(f_{X|q} + f_q) \circ f_{e_i}$ can accelerate model convergence, thereby reducing overfitting, especially when the model*

*has far more parameters than training data. This ultimately improves model generalization [16].*

**Theorem 3.1** can be proven based on the three common similarity functions of existing KGQE models. The detailed proofs are provided in **Appendix A1.2**. Here, we present the specific form of $(f_{X|q} + f_q) \circ f_{e_i}$ under each of the three similarity functions:

$$(f_{X|q} + f_q) \circ f_{e_i} = \|f_{e_i} - (f_{X|q} + f_q)\|_1, \quad (8)$$

$$(f_{X|q} + f_q) \circ f_{e_i} = \mathcal{D}_{KL}[P(\alpha_{e_i}, \beta_{e_i}) | P(\alpha_q + \alpha_{X|q}, \beta_q + \beta_{X|q})], \quad (9)$$

$$(f_{X|q} + f_q) \circ f_{e_i} = d_{out}(f_{e_i}, f_q + f_{X|q}) + \lambda d_{in}(f_{e_i}, f_q + f_{X|q}), \quad (10)$$

Eqs. (8), (9), and (10) are used to replace Eq. (7) when $\circ$ is a distance function based on $L_1$ norm [4, 10, 18, 31], KL divergence based on Beta distributions [32], and a distance function based on a cone space [44], respectively, where $P(\alpha, \beta)$ is the probability density function of a Beta distribution, $d_{in}(\cdot)$ and $d_{out}(\cdot)$ are used to measure the external and internal distances between entities and queries, respectively, and $\lambda$ is a fixed parameter. In addition, $\alpha_{e_i}, \alpha_q, \alpha_{X|q}, \beta_{e_i}, \beta_q, \beta_{X|q} \in \mathbb{R}^{1 \times \frac{F_2}{2}}$ satisfy $f_{e_i} = \alpha_{e_i} \| \beta_{e_i}, f_q = \alpha_q \| \beta_q, f_{X|q} = \alpha_{X|q} \| \beta_{X|q}$, where $\|$ is a column-wise vector concatenation operation.

## 3.4 Adaptive Normalization Component

In Theorem 3.1, we assume that $f_q + f_{X|q}$ satisfies a specific constraint on the value of $f_q$ (such as a cone boundary [44] or clamp constraint [10, 32]). To meet this assumption, we apply an adaptive normalization component $\mathcal{A}(\cdot)$ for $f_q + f_{X|q}$ in Theorem 3.1.

For a clamp constraint, $f_q + f_{X|q} = \{x^{(i)}\}_{i=1}^{F_2}$ can be converted by Eq. (11):

$$\mathcal{A}(f_q + f_{X|q}) = \{\max(\min(x^{(i)}, \mu_{max}), \mu_{min})\}_{i=1}^{F_2}, \quad (11)$$

where, $[\mu_{min}, \mu_{max}]$ are the boundary of a specific clamp constraint. When $f_q + f_{X|q}$ are defined in the real number, Beta distribution, and fuzzy reasoning space, $[\mu_{min}, \mu_{max}]$ are represented as $(-\infty, +\infty)$, $[0.05, +\infty)$, and $[0, 1]$, respectively.

When $f_q + f_{X|q}$ satisfies the cone boundary, $\mathcal{A}(\cdot)$ is set to Eq. (12):

$$\mathcal{A}(f_q + f_{X|q}) = \pi \tanh(\alpha_q + \alpha_{X|q}) \| [\pi + \pi \tanh 2(\beta_q + \beta_{X|q})], \quad (12)$$

where $\alpha_q, \alpha_{X|q}, \beta_q, \beta_{X|q} \in \mathbb{R}^{1 \times \frac{F_2}{2}}$ satisfy $f_q + f_{X|q} = (\alpha_q + \alpha_{X|q}) \| (\beta_q + \beta_{X|q})$.

## 3.5 Training and Reasoning

After selecting a suitable normalization function from Eqs. (11) and (12), the query embedding of $q$ can be defined as $f^* = \mathcal{A}(f_q + f_{X|q})$. Then, the loss function can be represented by Eq. (13):

$$\mathcal{L}(q, X, e^+, \{e_i^-\}_{i=1}^k) = -\log \sigma(\gamma - f^* \circ f_{e^+}) - \frac{1}{k} \sum_{i=1}^k \log \sigma(f^* \circ f_{e_i^-} - \gamma), \quad (13)$$

where $\sigma(\cdot)$ is a sigmoid function, $\gamma > 0$ is a fixed margin, $e^+, e_i^- \in \mathcal{E}$ are a positive entity and the $i$-th random sampled negative target entity of $q$, respectively. $f_{e^+}$ and $f_{e_i^-}$ are the embeddings of $e^+$ and $e_i^-$, respectively. For a specific $QE(\cdot)$, $f^*$, $f_{e^+}$, and $f_{e_i^-}$ correspond to $f_q + f_X$, $f_{e_i}$, and $f_{e_i}$ in Eq. (8) (or Eq. (9) or Eq. (10)), respectively. The detailed training process is provided in **Appendix A1.3**.

| Dataset | Entities | Relations | Training Queries | validating Queries | Testing Queries |
|---------|----------|-----------|------------------|--------------------|-----------------|
| FB15k-237 | 14,505 | 474 | 748,445 | 85,094 | 87,804 |
| FB15k | 14,951 | 2,690 | 1,368,550 | 163,078 | 170,990 |
| NELL-995 | 63,361 | 400 | 539,910 | 68,910 | 69,021 |

Table 1: Element statistics of the experimental datasets.

When reasoning the target entity of $q$, we calculate the scores of $q$ and each entity $e \in \mathcal{E}$ using $\log \sigma(\gamma - f^* \circ f_e)$, and select the highest-scoring entity as the inference result for $q$. It is important to note that we need to select a specific KGQE model according to Section 2.2 to implement this reasoning process.

## 4 Experiments

In this section, we assess the effectiveness of our proposed QIPP by discussing the following five research questions: **RQ1**. Does QIPP provide performance gains in both end-to-end and iterative KGQE frameworks, and does it outperform existing QPL plugins? **RQ2**. Does the construction of code-like instructions and the pre-trained BERT's awareness of instruction context contribute to QIPP's query pattern learning? **RQ3**. Can the pattern injection mechanism with a compressed optimization boundary improve QIPP's performance? **RQ4**. Is the adaptive normalization component beneficial for QIPP? **RQ5**. Does the scale of the instruction encoder influence QIPP's performance?

## 4.1 Dataset, Baseline, and Evaluation Metric

We conduct experiments using three widely used KG datasets: **FB15k-237** [9], **FB15k** [33], and **NELL995** [40], pre-processed by [32]. These datasets include nine positive query types (1p, 2p, 3p, 2i, 3i, ip, pi, 2u, up) as well as five types of queries with negative operations (2in, 3in, inp, pin, pni). To evaluate the method's generalization [4, 10, 31, 32, 44], we train a KGQE model on 5 types of query (1p, 2p, 3p, 2i, 3i) and test it on all 14 query types. The statistics for the three datasets are shown in Table 1. Refer to **Appendix A2.1** for details on data splitting.

We verify the effectiveness of QIPP by deploying it on eight basic QE models, including six end-to-end KGQE models (**GQE** [18], **Q2B** [31], **BetaE** [32], **ConE** [44], **MLP** [4], and **FuzzQE** [10]) and two iterative KGQE models (**CQD-Beam** [5] and **QTO** [7]). In addition, two QPL plugins (**TEMP** [19] and **CaQR** [22]) are used to compared with QIPP. The descriptions of baselines are provided in **Appendix A2.2**.

Given a test FOL query $q$ and its target entity set $\mathcal{E}_{target}$, let $\mathcal{E}_{pred} = \{e_1, e_2, ..., e_p\}$ be the entities sorted by the predicted scores, we use Mean Reciprocal Rank (MRR) to evaluate the above baselines and our proposed method:

$$MRR(q) = \frac{1}{p} \sum_{i=1}^{p} \frac{\mathbb{I}(e_i, \mathcal{E}_{target})}{rank(e_i)}, \mathbb{I}(e_i, \mathcal{E}_{target}) = \begin{cases} 1, e_i \in \mathcal{E}_{target} \\ 0, otherwise \end{cases}.$$

## 4.2 Experimental Setting

In our experiments, we use "*bert-base-cased*" as the basic PLM. Then, we train each model on two NVIDIA A100 GPUs with 95 GB memory. The layer number of the instruction encoder defaults to 1 and the number of attention heads $H$ is set to 4. The training

parameters for deploying QIPP on different basic KGQE models are provided as follows:

When QIPP is deployed for training on the end-to-end KGQE methods, we uniformly set the batch size to 512 and the number of negative entity samples $k = 128$. The remaining parameters adapted to different end-to-end KGQE methods are shown in Table A2 at **Appendix A2.2**.

For iterative KGQE methods, we set the batch size to 1000, the total epochs to 100, and the learning rate $\eta = 0.1$ to train a ComplEx model with QIPP. The remaining T-norm framework is executed according to the original parameters of each iterative KGQE model.

## 4.3 Main Experimental Analysis (RQ1)

Table 2 and Figure 4 show the enhancement of QIPP on different basic KGQE models and its comparison with different QPL plugins, respectively. We conduct specific analyses based on **RQ1**.

**Analysis of end-to-end KGQE models**. As shown in Table 2, our plugin improves the performance of most end-to-end basic KGQE models. For example, QIPP provides the most significant improvement for GQE, with an enhancement of over 40%. This is because GQE, the most basic KGQE model, employs the simplest methods to model the logical operations in FOL queries, which ignores many logical semantics. By effectively capturing query patterns, QIPP enables GQE to compensate for its limitation in learning logical semantics, thereby significantly enhancing GQE's reasoning performance.

For end-to-end KGQE models that capture more complete logical semantics (such as ConE), the enhancement effect of QIPP is less pronounced than its impact on GQE. This is because these KGQE models have built complex encoding components for various logical operations, partially balancing the gains provided by query patterns injected by QIPP. In addition, we find that on FB15k, the improvement of QIPP on end-to-end KGQE models based on geometry learning, such as Q2B and ConE, is insignificant. This may be due to FB15k containing more relations than the other two datasets, which significantly increases the difficulty for QIPP in learning semantic across different relations in various query patterns.

**Analysis of iterative KGQE models**. As shown in Table 2, QIPP significantly improves the inference performance of CQD-Beam and QTO on queries with unknown types (*e.g.*, 'ip', 'pi', '2u', and 'up'). Unlike being deployed in a end-to-end KGQE model to learn all available query patterns, QIPP only needs to provide 1p queries' patterns for iterative KGQE models. This allows a iterative KGQE model to focus more on obtaining entity transformations for each atomic query, improving training efficiency and enhancing the model's generalization with the assistance of a T-norm inference framework.

**Analysis of QPL plugins**. As shown in Figure 4 (detail results are provided in Table A4), QIPP consistently outperforms the other two QPL plugins across various basic KGQE models in most scenarios. Considering that TEMP performs random type sampling on each entity, it cannot ensure that the selected type labels accurately reflect the entity's pattern context within a specific query, leading to its overall poor performance. Although CaQR adds structural context for random type sampling to globally constrain query patterns, the sparse initialization of structural context limits its

| | Method | $\text{Avg}_{Pos}$ (gain) | $\text{Avg}_{Neg}$ (gain) | 1p | 2p | 3p | 2i | 3i | ip | pi | 2u | up | 2in | 3in | inp | pin | pni |
|---|---|---|---|---|---|---|---|---|---|---|---|---|---|---|---|---|---|---|
| | **FB15k-237** | | | | | | | | | | | | | | | | | |
| E2E. | GQE [18] | 0.163 | - | 0.350 | 0.072 | 0.053 | 0.233 | 0.346 | 0.107 | 0.165 | 0.082 | 0.057 | - | - | - | - | - |
| | +QIPP | **0.229** (+40.55%) | - | **0.44** | **0.120** | **0.101** | **0.341** | **0.476** | **0.144** | **0.208** | **0.137** | **0.092** | - | - | - | - | - |
| | Q2B [31] | 0.201 | - | 0.406 | 0.094 | 0.068 | 0.295 | 0.423 | 0.126 | 0.212 | 0.113 | 0.076 | - | - | - | - | - |
| | +QIPP | **0.222** (+10.03%) | - | **0.433** | **0.104** | **0.091** | **0.330** | **0.468** | **0.132** | **0.218** | **0.131** | **0.087** | - | - | - | - | - |
| | BetaE [32] | 0.208 | 0.054 | 0.392 | 0.107 | 0.098 | 0.290 | 0.425 | 0.119 | 0.221 | 0.126 | 0.097 | 0.051 | 0.079 | 0.074 | 0.035 | 0.034 |
| | +QIPP | **0.233** (+11.63%) | **0.056** (+2.93%) | **0.419** | **0.128** | **0.109** | **0.324** | **0.487** | **0.131** | **0.251** | **0.141** | **0.103** | **0.054** | 0.079 | **0.077** | **0.038** | 0.033 |
| | ConE [44] | 0.232 | 0.058 | 0.424 | 0.126 | 0.108 | 0.325 | 0.468 | 0.136 | 0.251 | 0.141 | 0.106 | 0.054 | 0.086 | 0.078 | 0.040 | 0.036 |
| | +QIPP | **0.238** (+2.68%) | **0.059** (+0.68%) | **0.430** | **0.129** | **0.109** | **0.337** | **0.482** | **0.139** | **0.256** | **0.150** | **0.109** | 0.054 | **0.089** | 0.078 | 0.037 | **0.038** |
| | MLP [4] | 0.226 | 0.068 | 0.427 | 0.124 | 0.106 | 0.317 | 0.439 | 0.149 | 0.242 | 0.137 | 0.097 | 0.064 | 0.106 | 0.080 | 0.046 | 0.044 |
| | +QIPP | **0.237** (+4.46%) | **0.069** (+0.69%) | **0.435** | **0.134** | 0.106 | **0.339** | **0.468** | 0.142 | **0.257** | **0.143** | **0.105** | **0.066** | 0.104 | **0.082** | 0.045 | **0.045** |
| | FuzzQE [10] | 0.244 | 0.077 | 0.443 | 0.138 | 0.102 | 0.330 | 0.474 | 0.188 | 0.263 | 0.156 | 0.108 | 0.085 | 0.116 | 0.078 | 0.052 | 0.059 |
| | +QIPP | **0.245** (+0.18%) | **0.078** (+1.03%) | 0.430 | 0.136 | **0.114** | **0.335** | 0.467 | **0.197** | **0.266** | 0.153 | 0.108 | **0.088** | 0.115 | 0.078 | **0.054** | 0.058 |
| Iter. | CQD-Beam [5] | 0.223 | - | 0.467 | 0.116 | 0.080 | 0.312 | 0.406 | 0.187 | 0.212 | 0.146 | 0.084 | - | - | - | - | - |
| | +QIPP | **0.246** (+10.05%) | - | 0.453 | **0.127** | 0.073 | **0.358** | **0.486** | **0.200** | **0.268** | **0.166** | 0.081 | - | - | - | - | - |
| | QTO [7] | 0.335 | 0.155 | 0.490 | 0.214 | 0.212 | 0.431 | 0.568 | 0.280 | 0.381 | 0.227 | 0.214 | 0.168 | 0.267 | 0.152 | 0.136 | 0.054 |
| | +QIPP | **0.347** (+3.55%) | **0.159** (+2.70%) | **0.526** | **0.236** | **0.224** | **0.437** | 0.538 | **0.300** | 0.377 | **0.261** | **0.225** | **0.185** | 0.259 | **0.159** | 0.133 | **0.062** |
| | **NELL995** | | | | | | | | | | | | | | | | | |
| E2E. | GQE [18] | 0.186 | - | 0.328 | 0.119 | 0.096 | 0.275 | 0.352 | 0.144 | 0.184 | 0.085 | 0.088 | - | - | - | - | - |
| | +QIPP | **0.271** (+46.14%) | - | **0.577** | **0.163** | **0.140** | **0.412** | **0.524** | **0.155** | **0.209** | **0.148** | **0.114** | - | - | - | - | - |
| | Q2B [31] | 0.229 | - | 0.422 | 0.140 | 0.112 | 0.333 | 0.445 | 0.168 | 0.224 | 0.113 | 0.103 | - | - | - | - | - |
| | +QIPP | **0.273** (+19.21%) | - | **0.580** | **0.147** | **0.138** | **0.413** | **0.536** | **0.169** | 0.219 | **0.137** | **0.120** | - | - | - | - | - |
| | BetaE [32] | 0.246 | 0.059 | 0.530 | 0.130 | 0.114 | 0.376 | 0.475 | 0.143 | 0.241 | 0.122 | 0.085 | 0.051 | 0.078 | 0.100 | 0.031 | 0.035 |
| | +QIPP | **0.263** (+6.91%) | **0.061** (+3.05%) | **0.563** | **0.148** | **0.131** | **0.386** | **0.502** | **0.158** | **0.246** | **0.130** | **0.101** | **0.058** | 0.076 | **0.108** | 0.029 | 0.033 |
| | ConE [44] | 0.271 | 0.064 | 0.531 | 0.161 | 0.139 | 0.400 | 0.508 | 0.175 | 0.263 | 0.153 | 0.113 | 0.057 | 0.081 | 0.108 | 0.035 | 0.039 |
| | +QIPP | **0.275** (+1.27%) | **0.065** (+1.56%) | **0.558** | **0.167** | **0.149** | 0.400 | 0.506 | 0.159 | **0.265** | **0.156** | **0.114** | 0.057 | 0.081 | **0.111** | **0.036** | **0.040** |
| | MLP [4] | 0.264 | 0.061 | 0.552 | 0.164 | 0.147 | 0.364 | 0.480 | 0.182 | 0.227 | 0.147 | 0.113 | 0.051 | 0.080 | 0.100 | 0.036 | 0.036 |
| | +QIPP | **0.285** (+7.82%) | **0.068** (+11.55%) | **0.581** | **0.180** | **0.156** | **0.405** | **0.503** | **0.192** | **0.265** | **0.156** | **0.124** | **0.061** | **0.085** | **0.114** | **0.039** | **0.039** |
| | FuzzQE [10] | 0.292 | 0.078 | 0.576 | 0.192 | 0.153 | 0.398 | 0.503 | 0.218 | 0.281 | 0.173 | 0.137 | 0.078 | 0.098 | 0.111 | 0.049 | 0.055 |
| | +QIPP | **0.300** (+2.66%) | 0.078 (+0.00%) | **0.588** | 0.189 | **0.158** | **0.418** | **0.527** | **0.219** | **0.287** | **0.178** | 0.137 | 0.078 | 0.098 | 0.110 | **0.051** | 0.054 |
| Iter. | CQD-Beam [5] | 0.286 | - | 0.604 | 0.206 | 0.116 | 0.393 | 0.466 | 0.239 | 0.254 | 0.175 | 0.122 | - | - | - | - | - |
| | +QIPP | **0.303** (+5.83%) | - | 0.598 | **0.214** | **0.131** | **0.409** | **0.499** | **0.253** | **0.287** | **0.193** | **0.141** | - | - | - | - | - |
| | QTO [7] | 0.328 | 0.129 | 0.607 | 0.241 | 0.216 | 0.425 | 0.506 | 0.265 | 0.313 | 0.204 | 0.179 | 0.138 | 0.179 | 0.169 | 0.099 | 0.059 |
| | +QIPP | **0.347** (+5.85%) | **0.152** (+17.70%) | **0.612** | **0.254** | **0.237** | **0.442** | 0.494 | **0.291** | **0.331** | **0.252** | **0.216** | **0.173** | **0.202** | **0.196** | **0.113** | **0.074** |

**Table 2: MRR of KGQE models with QIPP. "E2E." and "Iter." represent end-to-end and iterative KGQE models, respectively. The bold font indicates that a KGQE model has higher MRR values after adding QIPP. Results of FB15k are provided in Table A3.**

**Figure 4: Comparison between QIPP and different QPL plugins. Since GQE and FB15k are not used in CaQR's experiments, we only present the basic models and datasets shown in the above figure.**

| Basic Model | GQE | | |
|---|---|---|---|
| Dataset | FB15k-237 | FB15k | NELL995 |
| Basic model | 0.163 | 0.280 | 0.186 |
| +QIPP | **0.229** | **0.416** | **0.271** |
| +QIPP w/o CLI | 0.214 | 0.401 | 0.267 |
| +QIPP w/o PT-BERT | 0.221 | 0.413 | 0.270 |

**Table 3: $\text{Avg}_{Pos}$ MRR of QIPP and its variants to verify the effectiveness of the code-like instruction encoding process.**

ability to learn query patterns effectively. In contrast, QIPP's use of code-like instructions avoids introducing additional noise, enabling more effective pattern learning. Furthermore, the BERT-based instruction encoder leverages the context-aware pre-trained BERT to better initialize the query patterns within the code-like instructions, allowing the model to fully utilize the query patterns to improve entity prediction.

## 4.4 Further Analysis

### 4.4.1 *Effectiveness of the code-like instruction encoding (RQ2)*. We validate the effectiveness of QIPP's code-like instruction encoding process on GQE, a basic KGQE model with the highest gain from QIPP. As shown in Table 3, we design two variants of QIPP for comparison: QIPP without Code-Like Instructions (CLIs), which uses the FOL format (refer to Table A1) instead of CLI, and QIPP

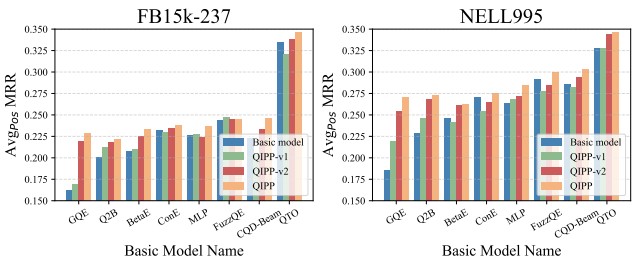

Figure 5: Comparison of QIPP with different variants. Results of FB15k are provided in Figure A1.

| Basic Model | BetaE | | |
|---|---|---|---|
| Dataset | FB15k-237 | FB15k | NELL995 |
| Basic model | 0.208 | 0.417 | 0.246 |
| +QIPP | **0.233** | **0.472** | **0.263** |
| +QIPP w/o ANC | NAN | NAN | NAN |
| Basic Model | ConE | | |
| Dataset | FB15k-237 | FB15k | NELL995 |
| Basic model | 0.232 | **0.497** | 0.271 |
| +QIPP | **0.238** | **0.497** | **0.275** |
| +QIPP w/o ANC | 0.209 | 0.465 | 0.233 |
| Basic Model | FuzzQE | | |
| Dataset | FB15k-237 | FB15k | NELL995 |
| Basic model | 0.244 | 0.459 | 0.292 |
| +QIPP | **0.245** | **0.506** | **0.300** |
| +QIPP w/o ANC | 0.234 | 0.454 | 0.268 |

"NAN" represents an out-of-bounds outlier.

Table 4: Avg$_{Pos}$ MRR of QIPP with/without ANC on basic KGQE models with special interval requirements.

without Pre-Trained (PT) BERT, which requires training a randomly initialized BERT module from scratch. Both variants demonstrate significantly weaker performance than QIPP. First, the intrinsic logical principles of queries expressed in the FOL format are less intuitive than those in code-like instructions. Unintelligible logical symbols in the FOL format can hinder the BERT-based instruction encoder's ability to extract query patterns, which limits subsequent instruction decoding and query pattern injection. Second, learning the logical semantics with limited code-like instructions from scratch is challenging. In contrast, pre-trained parameters can provide a higher-quality parameter space to guide QIPP in learning query patterns. In addition, because QIPP employs only one BERT encoding layer, the cost of learning code-like instructions from scratch is relatively low, resulting in better performance of QIPP w/o PT-BERT compared to QIPP w/o CLI.

*4.4.2 **Effectiveness of the query pattern injector (RQ3)**.* To demonstrate the effectiveness of the pattern injection mechanism, we design two variants of QIPP for experiments. Variant 1 (QIPP-v1) only optimizes the output of the instruction decoder, e.i., $f_q + f_{X|q}$ in Eq. (11) or Eq. (12) is replaced by $f_{X|q}$ in Eq. (4); Variant 2 (QIPP-v2) follows Eq. (7) and optimizes $f_q$ and $f_{X|q}$ separately, without using a compressed optimization boundary. Figure 5 presents the experimental results of QIPP and its variants. Among them, QIPP-v1 performs the worst, even lower than the basic QE model, because

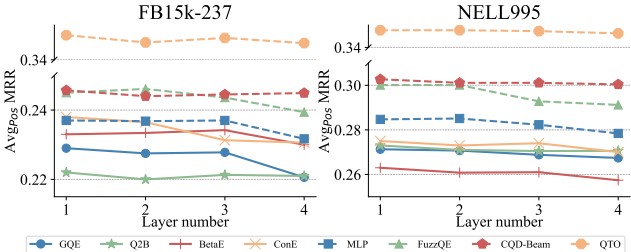

Figure 6: Avg$_{Pos}$ MRR of QIPP with the instruction encoder containing different layers. Results of FB15k are provided in Figure A2.

it ignores the micro logic semantics of queries (typically learned through geometric or probabilistic modules in QE models), making it challenging to achieve fine-grained inference of complex queries relying solely on the query patterns in code-like instructions. QIPP-v2 performs better than QIPP-v1 because it considers both the query's micro logic semantics and pattern information. However, without using compressed optimization boundaries, QIPP-v2 struggles to fit query patterns in a vast parameter space, making its performance inferior to QIPP.

*4.4.3 **Effectiveness of the adaptive normalization component (RQ4)**.* Table 4 shows the performance of QIPP with/without the adaptive normalization component (ANC). Obviously, ANC plays a crucial role in QIPP. Without ANC, the parsed query patterns are difficult to adapt to the basic QE models that require specific interval constraints, making it difficult to match the final query embeddings to the correct target entities. This can even lead to parameter out-of-bounds issues. For example, as shown in Table 4, when BetaE uses QIPP without ANC, the constructed query embeddings fail to meet the parameter requirements of the Beta distribution and result in NAN values.

*4.4.4 **Analysis of the Scale of the Instruction Encoder (RQ5)**.* As shown in Figure 6, we evaluate the performance of the BERT-based instruction encoder on QIPP with 1 to 4 layers. We find that as the number of layers increases, the gains of the instruction encoder on QIPP are not significant, and the performance of some KGQE models even tends to decline. This is because BERT layers capture instruction context at different scales [21]. Shallow layers focus on mining information at the level of sentence structure and variable phrases, which are diluted at the deep layers. In contrast, deep layers tend to obtain more abstract semantic representations for text matching. Since the task of QIPP is to obtain the logical pattern of the query, including the semantics of variables and the logical relations among hierarchical tuples, it aligns more closely with the information provided by BERT's shallow layers.

## 4.5 Case Study

To visually demonstrate the effectiveness of QIPP, we randomly selecte 100 queries from the NELL995 dataset for embedding visualization. For ease of presentation, these queries only contain one answer entity. Figure 7 visualizes the query embeddings obtained by Q2B after deploying TEMP, CaQR, and QIPP, respectively.

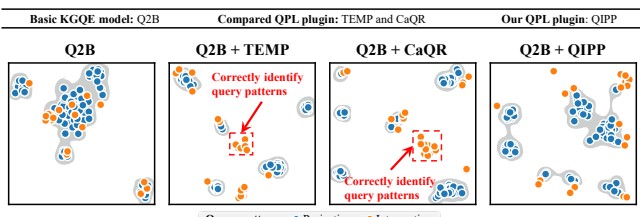

(a) Visualization of query embeddings colored according to query patterns. **The gray shadows** are the clustering interval of the queries with the projection pattern. The fewer intersection pattern queries contained within the shadows, the better the model can recognize the query patterns.

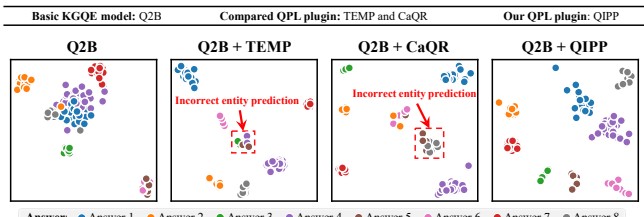

(b) Visualization of query embeddings colored according to answers (correct target entities of the queries). The closer the queries with the same answer are, the more accurate the model's prediction of the queries' answers.

**Figure 7: T-SNE visualization of query embeddings obtained from Q2B [31] before and after deploying TEMP [19], CaQR [22], and our proposed QIPP. We annotate these embeddings from the perspectives of (a) query pattern and (b) query answer.**

Specifically, due to the interference of randomly assigned redundant entity types, both TEMP and CaQR suffer from pattern-entity alignment bias, where they can correctly recognize the pattern of the queries in the red dashed box in Figure 7(a), but fail to predict the corresponding answer for the queries. In contrast, our QIPP, through code-like instructions and the PLM-based encoder-decoder framework, can correctly predict both the answer and pattern of the queries simultaneously.

## 5 Related Work

This section reviews the research of Knowledge Graph Reasoning (KGR), with a focus on the study of Knowledge Graph Complex Query Answering (KGCQA) tasks.

Early KGR methods focus on processing logical queries over visible KGs, using first-order logic with conjunction and existential quantization operators to retrieve KG subsets that match the query [1]. However, when a KG contains incomplete facts, this method is difficult to infer correct answers that are not visible. To address this issue, various studies have applied Knowledge Graph Embedding (KGE) [9, 34] to map entities and relations in incomplete KGs to a joint low-dimensional space, leveraging semantic similarity to predict unseen correct answers. In addition, some studies [12, 35, 40] have integrated reinforcement learning with search strategies to enhance the performance of the model on multi-hop queries.

To further enable models to handle queries containing complex logical operation combinations, researchers propose Knowledge Graph Query Embedding (KGQE) methods [11, 27, 39, 42] based on the KGE framework. These methods iteratively encode the entity-relation projection of each atomic query within complex queries. The GQE model proposed by William et al. [18] successfully implements complex query embeddings on a KGE framework for the first time. Following this, various GQE-based studies introduce additional query modeling techniques to obtain richer query logic semantics. For example, the Q2B model proposed by Ren et al. [31] uses box embeddings to represent a query as a hyper-rectangle and defines the predicted results as entities inside the rectangle. BetaE [32] models entities and queries as beta distributions, which enable flexible implementation of complex logical operations such as negation. ConE [44] models a query as a sector-cone space and regard the entities covered within the space as the prediction results. In addition, other studies have applied multi-layer perceptrons [4], fuzzy reasoning [10], eventuality KGs [6], hyper-relational KGs [2], and the Transformer framework [25, 26, 38, 41] to enhance KGQE models.

In contrast to the above methods that complicate the KGE framework, some studies aim to use traditional symbolic reasoning logic to guide the KGE framework in completing KGCQA tasks [5, 45]. Most of these methods employ a T-norm-based inference framework [17, 23, 24] to gradually infer the target entity step by step. For example, Arakelyan et al. propose the CQD model [5], which incorporates a Beam search strategy into the ComplEx model [34]. Wang et al. [36] better constrain the inference process of the T-norm framework by generating variable embeddings for intermediate answers. The QTO model proposed by Bai et al. [7] effectively reduces the time complexity of the T-norm inference framework by using a forward-backward propagation method on tree-like query graphs.

Given the challenge of exhaustively extracting all types of complex queries from KGs, recent studies have focused on constructing more generalizable Query Pattern Learning (QPL) frameworks to improve the model predictions for unknown queries. Methods like TEMP [19] and CaQR [22] typically use entity types or relation contexts as the basis for QPL, which allows the model to recognize similar pattern information in unknown queries. However, due to the randomness of type sampling and relation context initialization, existing QPL frameworks may introduce unnecessary external noise that can interfere with the model's ability to learn query patterns. To address this problem, our proposed QPL framework, QIPP, leverages code-like instructions and the Transformer framework to more efficiently induce query patterns, thereby enhancing the model's ability to utilize these query patterns for entity prediction.

## 6 Conclusion

This paper proposes a novel a plug-and-play Query Instruction Parsing Plugin (QIPP) for complex logic query answering tasks, which aims to address the limitations in the inference ability of KGQE models caused by pattern-entity alignment bias. QIPP consists of an encoder-decoder for code-like query instructions, a query pattern injection mechanism based on compressed optimization boundaries, and an adaptive normalization component. Theoretical proofs and extensive experimental analyses demonstrate QIPP's enhancement of eight basic KGQE models, its superiority over existing two query pattern learning methods, and the effectiveness of each component in QIPP.

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

# A1 Methodology Supplementary Materials

## A1.1 Code-like Instruction

Table A1 shows the 14 types of FOL queries and their corresponding code-like instruction formats used in our experiments.

| Type | FOL query format | Code-like instruction format |
|---|---|---|
| 1p | $\mathcal{E}_? : r_1(\{e_1\}, \mathcal{E}_?)$ | $(e_1, (r_1))$ |
| 2p | $\mathcal{E}_?.\exists \mathcal{E}_1 : r_1(\{e_1\}, \mathcal{E}_1)$ $\land r_2(\mathcal{E}_1, \mathcal{E}_?)$ | $(e_1, (r_1, r_2))$ |
| 3p | $\mathcal{E}_?.\exists \mathcal{E}_1, \mathcal{E}_2 : r_1(\{e_1\}, \mathcal{E}_1)$ $\land r_2(\mathcal{E}_1, \mathcal{E}_2) \land r_3(\mathcal{E}_2, \mathcal{E}_?)$ | $(e_1, (r_1, r_2, r_3))$ |
| 2i | $\mathcal{E}_? : r_1(\{e_1\}, \mathcal{E}_?)$ $\land r_2(\{e_2\}, \mathcal{E}_?)$ | $((e_1, (r_1)), (e_2, (r_2)))$ |
| 2in | $\mathcal{E}_? : r_1(\{e_1\}, \mathcal{E}_?)$ $\land \neg r_2(\{e_2\}, \mathcal{E}_?)$ | $((e_1, (r_1)), (e_2, (r_2, \text{negative})))$ |
| 3i | $\mathcal{E}_? : r_1(\{e_1\}, \mathcal{E}_?)$ $\land r_2(\{e_2\}, \mathcal{E}_?) \land r_3(\{e_3\}, \mathcal{E}_?)$ | $((e_1, (r_1)), (e_2, (r_2)), (e_3, (r_3)))$ |
| 3in | $\mathcal{E}_? : r_1(\{e_1\}, \mathcal{E}_?)$ $\land r_2(\{e_2\}, \mathcal{E}_?) \land \neg r_3(\{e_3\}, \mathcal{E}_?)$ | $((e_1, (r_1)), (e_2, (r_2)), (e_3, (r_3, \text{negative})))$ |
| ip | $\mathcal{E}_?.\exists \mathcal{E}_1 : r_1(\{e_1\}, \mathcal{E}_1)$ $\land r_2(\{e_2\}, \mathcal{E}_1) \land r_3(\mathcal{E}_1, \mathcal{E}_?)$ | $(((e_1, (r_1)), (e_2, (r_2))), (r_3))$ |
| inp | $\mathcal{E}_?.\exists \mathcal{E}_1 : r_1(\{e_1\}, \mathcal{E}_1)$ $\land \neg r_2(\{e_2\}, \mathcal{E}_1) \land r_3(\mathcal{E}_1, \mathcal{E}_?)$ | $(((e_1, (r_1)), (e_2, (r_2, \text{negative}))), (r_3))$ |
| pi | $\mathcal{E}_?.\exists \mathcal{E}_1 : r_1(\{e_1\}, \mathcal{E}_1)$ $\land r_2(\mathcal{E}_1, \mathcal{E}_?) \land r_3(\{e_2\}, \mathcal{E}_?)$ | $((e_1, (r_1, r_2)), (e_2, (r_3)))$ |
| pin | $\mathcal{E}_?.\exists \mathcal{E}_1 : r_1(\{e_1\}, \mathcal{E}_1)$ $\land r_2(\mathcal{E}_1, \mathcal{E}_?) \land \neg r_3(\{e_2\}, \mathcal{E}_?)$ | $((e_1, (r_1, r_2)), (e_2, (r_3, \text{negative})))$ |
| pni | $\mathcal{E}_?.\exists \mathcal{E}_1 : r_1(\{e_1\}, \mathcal{E}_1)$ $\land \neg r_2(\mathcal{E}_1, \mathcal{E}_?) \land r_3(\{e_2\}, \mathcal{E}_?)$ | $((e_1, (r_1, r_2, \text{negative})), (e_2, (r_3)))$ |
| 2u | $\mathcal{E}_? : r_1(\{e_1\}, \mathcal{E}_?)$ $\lor r_2(\{e_2\}, \mathcal{E}_?)$ | $(e_1, (r_1)), (e_2, (r_2))$ |
| up | $\mathcal{E}_?.\exists \mathcal{E}_1 : (r_1(\{e_1\}, \mathcal{E}_1)$ $\lor r_2(\{e_2\}, \mathcal{E}_1)) \land r_3(\mathcal{E}_1, \mathcal{E}_?)$ | $(e_1, (r_1, r_3)), (e_2, (r_2, r_3))$ |

**Table A1: Code-like instruction format of 14 query types. "p", "i", "u", and "n" represent the logic operation of projection, intersection/conjunction, union/disjunction, and negation, respectively. $e_1, e_2, e_3, r_1, r_2, r_3$ in the "Code-like instruction format" column represent the textual annotations of corresponding entities and relations in a specific FOL query.**

## A1.2 Proofs of Theorem 1

In this section, we provide detailed proofs of **Theorem 3.1** from the perspective of three kinds of similarity functions.

PROOF A1.1. When $\circ$ is a distance function based on $L_1$ norm [4, 10, 18, 31], Eq. (7) can be rewritten as:

$$\min (f_{X|q} \circ f_{e_i} + f_q \circ f_{e_i})$$
$$= \min (\|f_{e_i} - f_{X|q}\|_1 + \|f_{e_i} - f_q\|_1)$$
$$\geq \min (\|2f_{e_i} - (f_{X|q} + f_q)\|_1)$$
$$\geq \min (\|f_{e_i} - (f_{X|q} + f_q)\|_1)$$
$$= \min [(f_{X|q} + f_q) \circ f_{e_i}],$$

which prove that the parameter domain of $(f_{X|q} + f_q) \circ f_{e_i}$ is more compact than that of $f_{X|q} \circ f_{e_i} + f_q \circ f_{e_i}$. □

When $\circ$ is the KL divergence based on Beta distributions [32], we provide the following definitions that can assist in proving:

*Definition A1.1.* The probability density function of a Beta distribution is $P(\alpha, \beta) = \frac{x^{\alpha-1}(1-x)^{\beta-1}}{B(\alpha,\beta)} \in [0, 1]$, where $\alpha, \beta > 0$, the normalization function $B(\alpha, \beta) = \int_0^1 x^{\alpha-1}(1-x)^{\beta-1}dx = \frac{\Gamma(\alpha)\Gamma(\beta)}{\Gamma(\alpha+\beta)}$, and a Gamma function $\Gamma(\alpha) = (\alpha - 1)!$.

*Definition A1.2.* The derivative of the logarithm of a Gamma function is defined as the Digamma function $\Psi(\alpha) = \frac{d\log\Gamma(\alpha)}{d\alpha} = \frac{d\Gamma(\alpha)}{\Gamma(\alpha)d\alpha}$.

PROOF A1.2. Let $\boldsymbol{\alpha}_{e_i}, \boldsymbol{\alpha}_q, \boldsymbol{\alpha}_{X|q}, \boldsymbol{\beta}_{e_i}, \boldsymbol{\beta}_q, \boldsymbol{\beta}_{X|q} \in \mathbb{R}^{1\times\frac{F_2}{2}}$ satisfy $f_{e_i} = \boldsymbol{\alpha}_{e_i}\|\boldsymbol{\beta}_{e_i}, f_q = \boldsymbol{\alpha}_q\|\boldsymbol{\beta}_q, f_{X|q} = \boldsymbol{\alpha}_{X|q}\|\boldsymbol{\beta}_{X|q}$, where $\|$ is a column-wise vector concatenation operation, and $\alpha_e^{(k)}, \beta_e^{(k)}, \alpha_q^{(k)}, \beta_q^{(k)}, \alpha_{X|q}^{(k)}, \beta_{X|q}^{(k)}$ are the $k$-th values of $\boldsymbol{\alpha}_{e_i}, \boldsymbol{\beta}_{e_i}, \boldsymbol{\alpha}_q, \boldsymbol{\beta}_q, \boldsymbol{\alpha}_{X|q}, \boldsymbol{\beta}_{X|q}$, respectively. Let's first discuss the construction of $f_q \circ f_{e_i}$ in the case of a single value:

$$\mathcal{D}_{KL}[P(\alpha_e^{(k)}, \beta_e^{(k)})|P(\alpha_q^{(k)}, \beta_q^{(k)})]$$
$$= \log \frac{B(\alpha_q^{(k)}, \beta_q^{(k)})}{B(\alpha_e^{(k)}, \beta_e^{(k)})} + \frac{\alpha_e^{(k)} - \alpha_q^{(k)}}{B(\alpha_e^{(k)}, \beta_e^{(k)})} \int_0^1 x^{\alpha_e^{(f)}-1}(1-x)^{\beta_e^{(f)}-1}\log x dx$$
$$+ \frac{\beta_e^{(k)} - \beta_q^{(k)}}{B(\alpha_e^{(k)}, \beta_e^{(k)})} \int_0^1 x^{\alpha_e^{(f)}-1}(1-x)^{\beta_e^{(f)}-1}\log(1-x)dx. \tag{A1}$$

Next, we take $\int_0^1 x^{\alpha_e^{(f)}-1}(1-x)^{\beta_e^{(f)}-1}\log x dx$ as an example to simplify Eq. (A1). According to **Definition A1.1**, it can be transformed into

$$\int_0^1 x^{\alpha_e^{(f)}-1}(1-x)^{\beta_e^{(f)}-1}\log x dx$$
$$= \int_0^1 \frac{dx^{\alpha_e^{(f)}-1}}{d\alpha_e^{(f)}}(1-x)^{\beta_e^{(f)}-1}\log x dx = \frac{\partial B(\alpha_e^{(f)}, \beta_e^{(f)})}{\partial \alpha_e^{(f)}}. \tag{A2}$$

Then, according to **Definition A1.2**, we have

$$\frac{\partial \log B(\alpha_e^{(f)}, \beta_e^{(f)})}{\partial \alpha_e^{(f)}}$$
$$= \frac{d\Gamma(\alpha_e^{(f)})}{\Gamma(\alpha_e^{(f)})d\alpha_e^{(f)}} - \frac{d\Gamma(\alpha_e^{(f)} + \beta_e^{(f)})}{\Gamma(\alpha_e^{(f)} + \beta_e^{(f)})d(\alpha_e^{(f)} + \beta_e^{(f)})} \tag{A3}$$
$$= \Psi(\alpha_e^{(f)}) - \Psi(\alpha_e^{(f)} + \beta_e^{(f)}) = \frac{1}{B(\alpha_e^{(f)}, \beta_e^{(f)})} \frac{\partial B(\alpha_e^{(f)}, \beta_e^{(f)})}{\partial \alpha_e^{(f)}}.$$

Combine Eqs. (A2) and (A3), we have

$$\int_0^1 x^{\alpha_e^{(f)}-1}(1-x)^{\beta_e^{(f)}-1}\log x dx$$
$$= B(\alpha_e^{(f)}, \beta_e^{(f)})[\Psi(\alpha_e^{(f)}) - \Psi(\alpha_e^{(f)} + \beta_e^{(f)})]. \tag{A4}$$

Similarly,

$$\int_0^1 x^{\alpha_e^{(f)}-1}(1-x)^{\beta_e^{(f)}-1}\log(1-x)dx$$
$$= B(\alpha_e^{(f)}, \beta_e^{(f)})[\Psi(\beta_e^{(f)}) - \Psi(\alpha_e^{(f)} + \beta_e^{(f)})]. \tag{A5}$$

Because $B(\alpha_q^{(k)}, \beta_q^{(k)})$ and $B(\alpha_e^{(k)}, \beta_e^{(k)})$ are normalization terms, based on Eqs. (A1)-(A5), we can obtain the following results:

$$f_{X|q} \circ f_{e_i} + f_q \circ f_{e_i}$$
$$= \mathcal{D}_{KL}[P(\boldsymbol{\alpha}_{e_i}, \boldsymbol{\beta}_{e_i})|P(\boldsymbol{\alpha}_q, \boldsymbol{\beta}_q)] + \mathcal{D}_{KL}[P(\boldsymbol{\alpha}_{e_i}, \boldsymbol{\beta}_{e_i})|P(\boldsymbol{\alpha}_{X|q}, \boldsymbol{\beta}_{X|q})]$$
$$\propto \sum_{k=1}^{\frac{F_2}{2}} [(2\alpha_e^{(k)} - \alpha_q^{(k)} - \alpha_{X|q}^{(k)})[\Psi(\alpha_e^{(f)}) - \Psi(\alpha_e^{(f)} + \beta_e^{(f)})]$$
$$+ (2\beta_e^{(k)} - \beta_q^{(k)} - \beta_{X|q}^{(k)})[\Psi(\beta_e^{(f)}) - \Psi(\alpha_e^{(f)} + \beta_e^{(f)})]].$$

Similarly,

$$(f_{X|q} + f_q) \circ f_{e_i}$$
$$= \mathcal{D}_{KL}[P(\boldsymbol{\alpha}_{e_i}, \boldsymbol{\beta}_{e_i}) | P(\boldsymbol{\alpha}_q + \boldsymbol{\alpha}_{X|q}, \boldsymbol{\beta}_q + \boldsymbol{\beta}_{X|q})]$$
$$\propto \sum_{k=1}^{\frac{F_2}{2}} [(\alpha_e^{(k)} - \alpha_q^{(k)} - \alpha_{X|q}^{(k)}) [\Psi(\alpha_e^{(f)}) - \Psi(\alpha_e^{(f)} + \beta_e^{(f)})]$$
$$+ (\beta_e^{(k)} - \beta_q^{(k)} - \beta_{X|q}^{(k)}) [\Psi(\beta_e^{(f)}) - \Psi(\alpha_e^{(f)} + \beta_e^{(f)})]].$$

Obviously, $f_{X|q} \circ f_{e_i} + f_q \circ f_{e_i} > (f_{X|q} + f_q) \circ f_{e_i}$ when $\alpha_e^{(k)}$, $\beta_e^{(k)}$, $\alpha_q^{(k)}$, $\beta_q^{(k)}$, $\alpha_{X|q}^{(k)}$, and $\beta_{X|q}^{(k)}$ are all positive, which prove that the parameter domain of $(f_{X|q} + f_q) \circ f_{e_i}$ is more compact than that of $f_{X|q} \circ f_{e_i} + f_q \circ f_{e_i}$. □

When $\circ$ is a distance function based on a cone space [44], we provide the following definition that can assist in proving:

*Definition A1.3.* According to [44], the query embedding $f_q = \boldsymbol{\alpha}_q \| \boldsymbol{\beta}_q$ is used to model a sector-cone space, where $\boldsymbol{\alpha}_q \in [-\pi, \pi)^{1 \times \frac{F_2}{2}}$ and $\boldsymbol{\beta}_q \in [0, 2\pi]^{1 \times \frac{F_2}{2}}$ are the axes and apertures of the cone space. The distance of $f_q$ and an entity embedding $f_{e_i} \in [-\pi, \pi)^{1 \times \frac{F_2}{2}}$ is defined as:

$$f_q \circ f_{e_i} = d_{out}(f_{e_i}, f_q) + \lambda d_{in}(f_{e_i}, f_q),$$
$$d_{out}(f_{e_i}, f_q) = \| \min [ | \sin \frac{f_{e_i} - I_q}{2} |, | \sin \frac{f_{e_i} - U_q}{2} | ] \|_1$$
$$d_{in}(f_{e_i}, f_q) = \| \min [ | \sin \frac{f_{e_i} - \boldsymbol{\alpha}_q}{2} |, | \sin \frac{\boldsymbol{\beta}_q}{2} | ] \|_1,$$

where $I_q = \frac{\boldsymbol{\alpha}_q - \boldsymbol{\beta}_q}{2}$, $U_q = \frac{\boldsymbol{\alpha}_q + \boldsymbol{\beta}_q}{2}$, and $\lambda \in (0, 1)$ is a fixed parameter.

PROOF A1.3. According to **Definition A1.3**, we can obtain $| \sin \frac{\boldsymbol{\beta}_q}{2} |$, $| \sin \frac{f_{e_i} - \boldsymbol{\alpha}_q}{2} |$, $| \sin \frac{f_{e_i} - I_q}{2} |$, $| \sin \frac{f_{e_i} - U_q}{2} | \in [0, 1]^{1 \times \frac{F_2}{2}}$. Therefore, $f_q \circ f_{e_i} + f_{X|q} \circ f_{e_i} \in [0, F_2(1 + \lambda)]$.

Similarly, $(f_q + f_{X|q}) \circ f_{e_i}$ ca be represented as:

$$(f_q + f_{X|q}) \circ f_{e_i} = d_{out}(f_{e_i}, f_q + f_{X|q}) + \lambda d_{in}(f_{e_i}, f_q + f_{X|q}),$$

where $(f_q + f_{X|q}) \circ f_{e_i} \in [0, \frac{F_2(1+\lambda)}{2}] \subseteq [0, F_2(1 + \lambda)]$. Therefore, the parameter domain of $(f_{X|q} + f_q) \circ f_{e_i}$ is more compact than that of $f_{X|q} \circ f_{e_i} + f_q \circ f_{e_i}$. □

## A1.3 Training Framework

In this section, we elaborate on the training framework of QIPP based on the types of KGQE models.

*A1.3.1 **Training on end-to-end KGQE models**.* As outlined in Section 2.2, we can directly deploy QIPP on an end-to-end KGQE model. Algorithm 1 shows the training framework of QIPP on end-to-end KGQE models.

We first initialize the trainable parameters of QIPP and a speffic KGQE model and set the training arguments including batch size $b$, loss margin $\gamma$, learning rate $\eta$, and max training step $s$. During each training step, we randomly select $b$ FOL queries without putting them back at Step 4. For each selected query $q$, we sample its positive and negative entities, construct its code-like instruction, and obtain its basic embedding in Steps 7, 8, and 9, respectively. Next, QIPP is used to obtain the query embedding $f^*$ containing query pattern information in Steps 10-12. In Steps 13-15, we obtain the traing

---

**Algorithm 1** Training QIPP on end-to-end KGQE models

**Input:** FOL query set $Q$; entity set $\mathcal{E}$; trainable parameters of $INST_\Theta(\cdot)$ $\Theta$; trainable parameters of $QE(\cdot)$ $\Omega$; trainable parameters of the instruction decoder $W_Q, W_K, W_V, W_O$; margin $\gamma$; learning rate $\eta$; max training step $s$; batch size $b$.
**Output:** optimized parameters $\Theta, \Omega, W_Q, W_K, W_V, W_O$.

1: $Q_{train} \leftarrow Q$; $step = 0$
2: **for** $step < s$ **do**
3:  Obtain $Q^* \subseteq Q_{train}$ that contains $b$ randomly selected FOL queries
4:  $\mathcal{L}_{total} = 0$
5:  **for** $q$ in $Q^*$ **do**
6:   Obtain a positive entity $e^+$ and $k$ negative eitities $\{e^+_{(i)}\}_{i=1}^k$ for $q$ from $\mathcal{E}$
7:   Obtain the code-like instruction $X$ of $q$
8:   Obtain the embedding $f_q$ of $q$ using $QE(\cdot)$
\* \* \* \* \* \* \* \* \* \* \* \* \* \* \* \* \* QIPP begin \* \* \* \* \* \* \* \* \* \* \* \* \* \* \* \*
9:   Obtain $X$ using Eq. (3)
10:   Obtain $f_{X|q}$ using Eq. (4)
11:   Obtain $f^*$ using Rq (11) or Eq.(12) according to $QE(\cdot)$
\* \* \* \* \* \* \* \* \* \* \* \* \* \* \* \* QIPP end \* \* \* \* \* \* \* \* \* \* \* \* \* \* \* \* \*
12:   Choose a distance function from Eqs. (8)-(11) according to $QE(\cdot)$
13:   Obtain a loss $\mathcal{L}$ by Eq. (13)
14:   $\mathcal{L}_{total} \leftarrow \mathcal{L}_{total} + \mathcal{L}$
15:  **end for**
16:  Optimize $\Theta, \Omega, W_Q, W_K, W_V, W_O$ using $\mathcal{L}_{total}$ with the Adam gradient descent method
17:  $step \leftarrow step + 1$
18:  $Q_{train} \leftarrow Q_{train} - Q^*$
19:  **if** $Q_{train} \in \varnothing$ **then**
20:   $Q_{train} \leftarrow Q$
21: **end for**
22: **return** $\Theta, \Omega, W_Q, W_K, W_V, W_O$

---

loss of query answering by calculating the embedding similarity between the given query and sampled positive/negative entities. We perform the Adam gradient descent operation on the overall loss of accumulating a batch of queries in Step 17 to optimize the trainable parameters in QIPP and a specific KGQE model. Finally, we transplant the optimized trainable parameters to the testing queries for evaluation.

*A1.3.2 **Training on iterative KGQE models**.* Unlike end-to-end KGQE model, iterative KGQE model only trains the ComplEx model [34] for atomic queries. Algorithm 2 presents the training process of deploying QIPP on ComplEx.

First, according to ComplEx, we obtain the embeddings of the anchor entity, relation, and target entity of an atomic query in complex space at Steps 7-10, where superscripts $(re)$ and $(im)$ represent the embeddings of the real and imaginary parts, respectively. Then, according to the process of obtaining query embeddings from ComplEx, we execute QIPP in Steps 13, 16, 20, and 23, respectively ($\odot$ is a Hadamard product operation), to calculate the loss function that is used to optimize the trainable parameters in QIPP and ComplEx.

---

**Algorithm 2** Training QIPP on ComplEx [34]

**Input:** Atomic query set $Q$; trainable parameters of $INST_{\Theta}(\cdot)$ $\Theta$; trainable parameters of the instruction decoder $W_Q, W_K, W_V, W_O$; trainable entity embeddings $E = \{(e_n^{(re)}, e_n^{(im)})\}_{n=1}^{N}$; trainable relation embeddings $R = \{(r_m^{(re)}, r_m^{(im)})\}_{m=1}^{M}$; learning rate $\eta$; max training step $s$; batch size $b$.

**Output:** optimized parameters $\Theta, E, R, W_Q, W_K, W_V, W_O$.

1: $Q_{train} \leftarrow Q$; $epoch = 0$
2: **for** $step < s$ **do**
3:     Obtain $Q^* \subseteq Q_{train}$ that contains $b$ randomly selected 1p queries
4:     $\mathcal{L}_{total} = 0$
5:     **for** $q$ in $Q^*$ **do**
6:         Obtain the target entity $e_p$, anchor entity $e_a$, and relation $r_m$ of $q$
7:         Obtain the embeddings $(e_t^{(re)}, e_t^{(im)})$, $(e_a^{(re)}, e_a^{(im)})$, and $(r_m^{(re)}, r_m^{(im)})$ of $e_p \in E, e_a \in E$, and $r_m \in R$, respectively.
\* \* \* \* \* \* \* \* \* \* \* \* \* \* \* QIPP begin \* \* \* \* \* \* \* \* \* \* \* \* \* \* \* \* \*
8:         Let $X = (e_a, (r_m))$ and obtain $X$ using Eq. (3)
9:         $f_q \leftarrow e_a^{(re)} \odot r_m^{(re)} - e_a^{(im)} \odot r_m^{(im)}$
10:        Obtain $f_{X|q}$ using Eq. (4) and set $f^* \leftarrow f_{X|q} + f_q$
11:        $\mathcal{L}_{total} \leftarrow -\log(sigmoid(e_t^{(re)}[f^*]^T)) + \mathcal{L}_{total}$
12:        $f_q \leftarrow e_a^{(im)} \odot r_m^{(re)} + e_a^{(re)} \odot r_m^{(im)}$
13:        Obtain $f_{X|q}$ using Eq. (4) and set $f^* \leftarrow f_{X|q} + f_q$
14:        $\mathcal{L}_{total} \leftarrow -\log(sigmoid(e_n^{(im)}[f^*]^T)) + \mathcal{L}_{total}$
15:        Let $X = (e_t, (inverse\_r_m))$ and obtain $X$ using Eq. (3)
16:        $f_q \leftarrow e_t^{(re)} \odot r_m^{(re)} + e_t^{(im)} \odot r_m^{(im)}$
17:        Obtain $f_{X|q}$ using Eq. (4) and set $f^* \leftarrow f_{X|q} + f_q$
18:        $\mathcal{L}_{total} \leftarrow -\log(sigmoid(e_t^{(re)}[f^*]^T)) + \mathcal{L}_{total}$
19:        $f_q \leftarrow e_t^{(im)} \odot r_m^{(re)} - e_t^{(re)} \odot r_m^{(im)}$
20:        Obtain $f_{X|q}$ using Eq. (4) and set $f^* \leftarrow f_{X|q} + f_q$
21:        $\mathcal{L}_{total} \leftarrow -\log(sigmoid(e_n^{(im)}[f^*]^T)) + \mathcal{L}_{total}$
\* \* \* \* \* \* \* \* \* \* \* \* \* \* \* \* QIPP end \* \* \* \* \* \* \* \* \* \* \* \* \* \* \* \*
22:     **end for**
23:     Optimize $\Theta, E, R, W_Q, W_K, W_V, W_O$ using $\mathcal{L}_{total}$ with the Adam gradient descent method
24:     $step \leftarrow step + 1$
25:     $Q_{train} \leftarrow Q_{train} - Q^*$
26:     **if** $Q_{train} \in \emptyset$ **then**
27:         $Q_{train} \leftarrow Q$
28: **end for**
29: **return** $\Theta, E, R, W_Q, W_K, W_V, W_O$

---

Finally, we attach a T-norm inference framework to the trained parameters to adapt to complex FOL query answering.

## A2 Experimental Supplementary Materials

### A2.1 Details of Data Splitting

To assess the model's predictive ability for unknown entities, we follow these specific steps [32] to obtain our experimental datasets[2].

---

[2]https://github.com/snap-stanford/KGReasoning

| KGQE method | Embedding size $F_2$ | Margin $\gamma$ | Learning rate $\eta$ | Batch size $b$ | Max training step $s$ |
|---|---|---|---|---|---|
| GQE | 800 | 24 | $1e^{-4}$ | 512 | 450,000 |
| Q2B | 400 | 24 | $1e^{-4}$ | 512 | 450,000 |
| BetaE | 400 | 60 | $1e^{-4}$ | 512 | 450,000 |
| ConE | 800 | 30 | $5e^{-5}$ | 512 | 300,000 |
| FuzzQE | 1,000 | 20 | $5e^{-4}$ | 512 | 450,000 |
| MLP | 800 | 24 | $1e^{-4}$ | 512 | 450,000 |
| ComplEx in CQD-Beam | 1,000 | - | 0.1 | 1,000 | 100,000 |
| ComplEx in QTO | 1,000 | - | 0.1 | 1,000 | 100,000 |

**Table A2: Training parameters of various basic KGQE methods.**

First, based on the standard spliting of each KG dataset, we obtain a training graph $\mathcal{G}_{train}$, a validating KG $\mathcal{G}_{valid}$, and a testing KG $\mathcal{G}_{test}$ without overlapping triplets. For each training query, we allow it to contain only positive entities from $\mathcal{G}_{train}$. For a validating/testing query, the positive target entities may be distributed across both $\mathcal{G}_{train}$ (visible) and $\mathcal{G}_{valid}/\mathcal{G}_{test}$ (invisible). Therefore, during the validating/testing process, we only score and rank the positive entities in $\mathcal{G}_{valid}/\mathcal{G}_{test}$ alongside all negative entities.

### A2.2 Descriptions of Basic KGQE Models

In this section, we introduce the basic KGQE models used in our experiments and provide their source codes and experimental parameter settings.

**GQE**[2] [18] is a typical end-to-end KGQE model. It embeds an FOL query using entities and relations in an Euclidean space, and then obtains the predicted target entities by calculating the $L_1$ distance between queries and entities

**Q2B**[2] [31] extends GQE into a hyper-rectangle space, which allows it to more accurately capture embedding offsets between queries and entities.

**BetaE**[2] [32] introduces the Beta distribution and probability calculation to achieve negative operations that GQE and Q2B cannot complete.

**ConE**[3] [44] models FOL queries as sector-cone intervals, making it the first geometry-based QE model to complete negative operations.

**MLP**[4] [4] improves the adaptability of GQE to conjunction and disjunction operations through several stacked linear layers, while implementing negation operation on GQE.

**FuzzQE**[5] [10] introduces the axiomatic system of logic into GQE, which uses fuzzy logic to simplify the encoding complexity of neural networks for conjunction, disjunction, and negation operations.

**CQD-Beam**[6] [5] uses ComplEx [34] to learn the transition between entities. For complex queries, CQD-Beam utilizes the T-norm theory and beam search method to achieve entity prediction, reducing the dependence of model training on query types and improving the generalization of the model.

---

[3]https://github.com/MIRALab-USTC/QE-ConE
[4]https://github.com/amayuelas/NNKGReasoning
[5]https://github.com/stasl0217/FuzzQE-code
[6]https://github.com/uclnlp/cqd

| | Method | Avg$_{Pos}$ (gain) | Avg$_{Neg}$ (gain) | 1p | 2p | 3p | 2i | 3i | ip | pi | 2u | up | 2in | 3in | inp | pin | pni |
|---|---|---|---|---|---|---|---|---|---|---|---|---|---|---|---|---|---|
| | | | | | | | | | | | | | FB15k | | | | |
| E2E. | GQE [18] | 0.280 | - | 0.546 | 0.153 | 0.108 | 0.397 | 0.514 | 0.191 | 0.276 | 0.221 | 0.116 | - | - | - | - | - |
| | +QIPP | **0.416** (+48.45%) | - | **0.725** | **0.268** | **0.213** | **0.583** | **0.688** | **0.245** | **0.361** | **0.415** | **0.236** | - | - | - | - | - |
| | Q2B [31] | 0.380 | - | 0.680 | 0.210 | 0.142 | 0.551 | 0.665 | 0.261 | 0.394 | 0.351 | 0.167 | - | - | - | - | - |
| | +QIPP | **0.381** (+0.32%) | - | **0.690** | 0.206 | 0.140 | 0.533 | 0.656 | **0.265** | **0.416** | **0.368** | 0.158 | - | - | - | - | - |
| | BetaE [32] | 0.417 | 0.118 | 0.651 | 0.257 | 0.247 | 0.558 | 0.665 | 0.281 | 0.439 | 0.401 | 0.252 | 0.143 | 0.147 | 0.115 | 0.065 | 0.124 |
| | +QIPP | **0.472** (+13.25%) | **0.120** (+1.01%) | **0.740** | **0.300** | **0.273** | **0.653** | **0.749** | 0.268 | **0.477** | **0.498** | **0.290** | **0.148** | 0.146 | **0.121** | 0.062 | 0.123 |
| | ConE [44] | 0.497 | 0.148 | 0.733 | 0.338 | 0.292 | 0.644 | 0.733 | 0.357 | 0.509 | 0.557 | 0.314 | 0.179 | 0.187 | 0.125 | 0.098 | 0.151 |
| | +QIPP | 0.497 (+0.00%) | **0.149** (+0.27%) | **0.784** | 0.323 | 0.290 | 0.641 | **0.739** | 0.35 | 0.481 | 0.555 | 0.314 | 0.178 | 0.176 | **0.131** | **0.101** | **0.156** |
| | MLP [4] | 0.439 | 0.139 | 0.671 | 0.312 | 0.272 | 0.571 | 0.669 | 0.339 | 0.457 | 0.380 | 0.280 | 0.160 | 0.173 | 0.132 | 0.083 | 0.146 |
| | +QIPP | **0.447** (+1.92%) | **0.142** (+2.02%) | **0.706** | **0.317** | 0.266 | **0.583** | **0.687** | 0.304 | **0.473** | **0.400** | **0.291** | **0.166** | **0.178** | **0.136** | **0.086** | 0.142 |
| | FuzzQE [10] | 0.459 | 0.140 | 0.772 | 0.331 | 0.253 | 0.587 | 0.690 | 0.323 | 0.479 | 0.401 | 0.294 | 0.180 | 0.167 | 0.134 | 0.088 | 0.134 |
| | +QIPP | **0.506** (+10.29%) | **0.141** (+0.14%) | **0.801** | **0.367** | **0.288** | **0.650** | **0.743** | **0.392** | **0.502** | **0.477** | **0.335** | 0.177 | 0.165 | **0.138** | 0.088 | **0.136** |
| Iter. | CQD-Beam [5] | 0.582 | - | 0.892 | 0.543 | 0.286 | 0.744 | 0.783 | 0.677 | 0.582 | 0.424 | 0.309 | - | - | - | - | - |
| | +QIPP | **0.644** (+10.61%) | - | 0.889 | **0.615** | **0.275** | **0.768** | **0.808** | **0.690** | **0.641** | **0.735** | **0.375** | - | - | - | - | - |
| | QTO [7] | 0.741 | 0.493 | 0.895 | 0.674 | 0.588 | 0.803 | 0.836 | 0.740 | 0.752 | 0.767 | 0.613 | 0.611 | 0.612 | 0.476 | 0.489 | 0.275 |
| | +QIPP | **0.763** (+2.99%) | **0.508** (+3.16%) | **0.904** | **0.708** | **0.609** | **0.822** | 0.835 | **0.768** | **0.771** | **0.803** | **0.648** | **0.626** | 0.607 | **0.528** | **0.509** | 0.271 |

**Table A3: MRR of KGQE models with QIPP on the FB15k dataset. "E2E." and "Iter." represent end-to-end and iterative KGQE models, respectively. The bold font indicates that a KGQE model has higher MRR values after adding QIPP.**

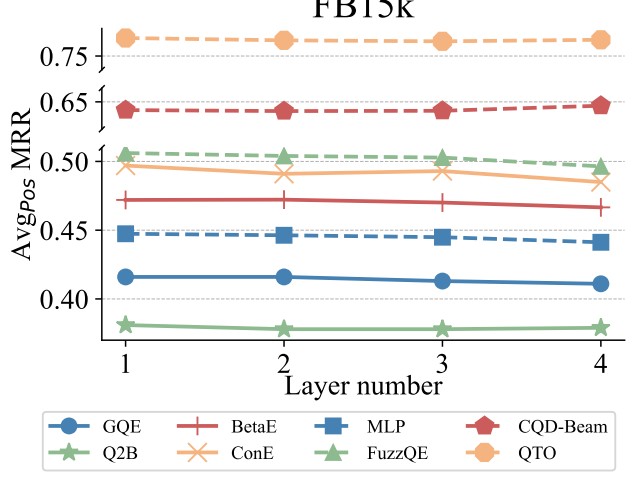

**Figure A1: Comparison of QIPP with different variants on FB15k**

**Figure A2: Avg$_{Pos}$ MRR of QIPP with the instruction encoder containing different layers on FB15k**

| Method | Avg. | 1p | 2p | 3p | 2i | 3i | ip | pi | 2u | up |
|---|---|---|---|---|---|---|---|---|---|---|
| | | | | FB15k-237 | | | | | | |
| GQE | 0.163 | 0.350 | 0.072 | 0.053 | 0.233 | 0.346 | 0.107 | 0.165 | 0.082 | 0.057 |
| +TEMP | 0.228 | 0.429 | **0.123** | **0.101** | **0.344** | **0.476** | **0.150** | **0.210** | 0.121 | **0.099** |
| +CaQR | - | - | - | - | - | - | - | - | - | - |
| +QIPP | **0.229** | **0.44** | 0.120 | **0.101** | 0.341 | **0.476** | 0.144 | 0.208 | **0.137** | 0.092 |
| Q2B | 0.201 | 0.406 | 0.094 | 0.068 | 0.295 | 0.423 | 0.126 | 0.212 | 0.113 | 0.076 |
| +TEMP | 0.220 | 0.409 | 0.110 | **0.092** | 0.337 | **0.482** | 0.123 | 0.214 | 0.124 | **0.091** |
| +CaQR | **0.222** | 0.426 | **0.108** | 0.090 | 0.324 | 0.464 | **0.134** | 0.217 | **0.145** | 0.086 |
| +QIPP | **0.222** | **0.433** | 0.104 | 0.091 | 0.330 | 0.468 | 0.132 | **0.218** | 0.131 | 0.087 |
| BetaE | 0.208 | 0.392 | 0.107 | 0.098 | 0.290 | 0.425 | 0.119 | 0.221 | 0.126 | 0.097 |
| +TEMP | 0.225 | 0.399 | 0.118 | 0.105 | **0.326** | 0.467 | 0.136 | 0.249 | 0.125 | 0.102 |
| +CaQR | 0.229 | 0.413 | 0.127 | 0.108 | 0.319 | 0.455 | **0.147** | 0.247 | **0.142** | **0.106** |
| +QIPP | **0.233** | **0.419** | **0.128** | **0.109** | 0.324 | **0.487** | 0.131 | **0.251** | 0.141 | 0.103 |
| | | | | NELL995 | | | | | | |
| GQE | 0.186 | 0.328 | 0.119 | 0.096 | 0.275 | 0.352 | 0.144 | 0.184 | 0.085 | 0.088 |
| +TEMP | 0.270 | 0.570 | **0.171** | **0.147** | 0.410 | 0.510 | **0.160** | **0.230** | 0.120 | 0.110 |
| +CaQR | - | - | - | - | - | - | - | - | - | - |
| +QIPP | **0.271** | **0.577** | 0.163 | 0.140 | **0.412** | **0.524** | 0.155 | 0.209 | **0.148** | **0.114** |
| Q2B | 0.229 | 0.422 | 0.140 | 0.112 | 0.333 | 0.445 | 0.168 | 0.224 | 0.113 | 0.103 |
| +TEMP | 0.264 | 0.562 | 0.150 | 0.129 | 0.408 | 0.520 | 0.160 | 0.211 | 0.142 | 0.094 |
| +CaQR | 0.269 | 0.571 | **0.160** | 0.134 | 0.379 | 0.512 | **0.175** | **0.242** | **0.148** | 0.104 |
| +QIPP | **0.273** | **0.580** | 0.147 | **0.138** | **0.413** | **0.536** | 0.169 | 0.219 | 0.137 | **0.120** |
| BetaE | 0.246 | 0.530 | 0.130 | 0.114 | 0.376 | 0.475 | 0.143 | 0.241 | 0.122 | 0.085 |
| +TEMP | 0.255 | 0.541 | 0.142 | 0.124 | 0.381 | 0.489 | 0.160 | 0.239 | 0.128 | 0.092 |
| +CaQR | 0.262 | 0.552 | **0.154** | **0.135** | 0.380 | 0.484 | **0.166** | **0.254** | **0.130** | **0.102** |
| +QIPP | **0.263** | **0.563** | 0.148 | 0.131 | **0.386** | **0.502** | 0.158 | 0.246 | **0.130** | 0.101 |

**Table A4: Comparison between QIPP and different QPL plugins.**

**QTO**[7] [7] optimizes the exponential search complex of CQD-Beam by a forward-backward propagation on the tree-like computation graphs.

**TEMP**[8] [19] is a QPL plugin that induces entity types into query pattern information, guiding a KGQE model to implement generalizable KGCQA tasks.

**CaQR**[9] [22] is a QPL plugin that further constrains the recognition boundaries of KGQE models for different types of queries by introducing structured relation context.

The training parameters of the above basic KGQE models are provided in Table A2.

[7]https://github.com/bys0318/QTO
[8]https://github.com/SXUNLP/QE-TEMP
[9]https://github.com/kjh9503/caqr

