# OpenReview forum: "Effective Instruction Parsing Plugin for Complex Logical Query Answering on Knowledge Graphs"
_ACM.org/TheWebConf/2025/Conference — WWW 2025 Poster_

### Official Review · Reviewer_KZrj · 2024-11-22

**Novelty:** 3
**Technical Quality:** 3

**Review:**

This paper focuses on the query pattern learning problem in complex logical query answering task. The authors identify the pattern-entity alignment bias problem in existing query pattern learning methods, which hampers KGQE performance. They propose a Query Instruction Parsing Plugin (QIPP) that uses pre-trained language models to derive query patterns from code-like query instructions, providing a new format to represent First-Order Logic (FOL) queries. QIPP includes a query-guided decoder and a query pattern injection mechanism to adapt and optimize these patterns for KGQE models.

Strengths:

- The paper is well-written and easy to follow, facilitating understanding of the concepts presented.
- The identification of the pattern-entity alignment bias problem might be interesting.
- The proposed method, QIPP, demonstrates improvements over existing baseline methods.

Weaknesses:

- The code-like instructions are almost same with the representation of query structures in Q2B and BetaE, the only difference is that the indexes of entities and relation names are replaced by text.
- The paper lacks a clear explanation of why the use of code-like instructions can effectively eliminate noise and redundant types. The reliance on a black-box pre-trained language model for encoding these instructions may still introduce unexplainable noise, undermining the claimed benefits.
- The motivation for addressing the pattern-entity alignment bias in query pattern learning is insufficiently substantiated. While the method improves baseline performance, its gains over other query-pattern learning approaches, such as TEMP and CaQR, are not particularly significant, as indicated in Figure 4. This weakens the evidence supporting the criticality of the pattern-entity alignment bias issue.
- In Section 4.5, the case study demonstrates that both TEMP and CaQR are capable of accurately identifying query patterns. This observation appears to contradict the conclusion presented in lines 574–577.

Thus, I think this paper lacks sufficiently compelling evidence to substantiate the significance of the pattern-entity alignment issue in query pattern learning.

**Questions:**

Q1: It would be good if the authors could provide a more detailed explanation of the advantages of employing code-like instructions and pre-trained language models in reducing noise and facilitating effective pattern learning.

Q2: In Figure 7 (a), the queries involving with intersection should also involve with projection, if I understand correctly. For example, 2i consists of two 1-hop queries. So, how to classify these queries into two types: projection and intersection?

Q3: In comparison with TEMP and CaQR, performance improvements of QIPP do not appear to be particularly substantial. Could you elaborate on any additional advantages that QIPP offers compared to TEMP and other query pattern learning methods?

**Reviewer Confidence:**

4: The reviewer is certain that the evaluation is correct and very familiar with the relevant literature

**Scope:**

4: The work is relevant to the Web and to the track, and is of broad interest to the community

---

### Official Review · Reviewer_JDZk · 2024-11-23

**Novelty:** 4
**Technical Quality:** 5

**Review:**

This paper proposes an instruction parsing plugin QIPP to enhance query graph embedding for knowledge graph query processing. QIPP transforms an input query into "code-like" instructions that represent the query with entity mentions and relationships extracted from the query, enclosed by parentheses to reflect their hierarchical/logical structure. The code-like instructions are encoded with a pre-trained language model to learn the query semantics, and then the embeddings are fused with the query graph embeddings with an attention mechanism (the "Instruction decoder"). The fused embedding and the query graph embedding further combined with adaptive normalization to produce the final query answer entity embedding estimation.

Experimental results show that QIPP can be combined with both end-to-end and iterative knowledge query graph embedding models to enhance their accuracy for knowledge graph query processing.

Pros:

1. The paper is well written in general.

2. The proposed plugin was combined with both end-to-end and iterative KGQE models and shown to improve their result accuracy, and the components of the proposed plugin are shown to be effective with ablation studies.

3. Theoretical analysis is presented to support the model design.

Cons:

1. The proposed QIPP model is somewhat incremental from existing plugin models [19, 22] to enhance query graph embedding. The existing works add entity types [19] and relation context [22] to query graph embeddings, while QIPP uses a pre-trained language model to encode the entity and relationship mentions from an input query to enrich the semantics of the query graph embeddings. The paper argues the importance of the structure of the code-like instructions. It would be good to add a model variant in the ablation study where the parentheses are removed from the code-like instructions and the instruction encoder encodes only the entity and relationship mentions.

2. Experimental setup can be improved. The current setup emphasizes that QIPP helps enhance different query graph embedding models which is okay. To show the advantage of QIPP, more results need to be presented to compare QIPP with the two existing plugins TEMP [19] and CaQR [22]. Figure 4 and Table A4 presented some results. However, it is unclear why only a subset of the base models were chosen for the comparison. Even with this chosen subset, the advantage of QIPP over TEMP and CaQR are marginal, and TEMP and CaQR are better for quite a few query types as shown in Table A4.

3. (Minor issue) Sentences like "Let q, g, and f_en be an FOL query q, the computation graph g of q, and the embedding of en..." read awkwardly. The paper could still use a proofread to polish the writing.

**Questions:**

See comments above.

**Reviewer Confidence:**

3: The reviewer is confident but not certain that the evaluation is correct

**Scope:**

3: The work is somewhat relevant to the Web and to the track, and is of narrow interest to a sub-community

---

### Official Review · Reviewer_Knr5 · 2024-12-01

**Novelty:** 5
**Technical Quality:** 5

**Review:**

The paper proposes QIPP, a novel Query Instruction Parsing Plugin for complex logic query answering tasks. QIPP uses code-like query instructions to solve the pattern-entity alignment bias problem which would limit the inference ability of KGQE models. In addition, QIPP uses an encoder-decoder, a query pattern injection mechanism based on compressed optimization boundaries, and an adaptive normalization component to make traditional QPL plugins adapt to both the end-to-end and iterative KGQE models. The approach seemingly enhances eight basic KGQE models and improves accuracy over existing query pattern learning methods. However the methodology section of the paper is somewhat complex, and I can't understand why such improvements could enhance the final results through simple formula derivation.

**Questions:**

Q1: The theoretical proof process is too complex to fully understand why QIPP could adapt to both the end-to-end and iterative KGQE models.
Q2: What are the advantages of code-like instruction compared to traditional FOL query, and how does it handle the pattern-entity alignment bias problem? I think it lacks direct comparative experiments between code-like instruction formats and other traditional FOL query formats.

**Reviewer Confidence:**

2: The reviewer is willing to defend the evaluation, but it is likely that the reviewer did not understand parts of the paper

**Scope:**

4: The work is relevant to the Web and to the track, and is of broad interest to the community

---

### Official Review · Reviewer_cwtW · 2024-12-02

**Novelty:** 4
**Technical Quality:** 4

**Review:**

Summary: This paper proposes an effective Query Instruction Parsing Plug-in (QIPP), which captures potential query patterns from code-like query instructions by leveraging the context-awareness ability of pre-trained language models (PLMs) to address the pattern-entity alignment bias issue in existing Query Pattern Learning (QPL) methods.

Strengths:
1. The issue of entity alignment bias is of practical value.
2. The authors have provided theoretical analysis for the proposed Query Pattern Injection module.

Weaknesses:
1. The motivation of the method is not clear enough. Instructions are powerful and mainly act on large language models. It is not clearly explained why the effect of the instructions in the paper is better than that of First-Order Logic (FOL) for query pattern learning task. Moreover, it is not explained why converting FOL into code-like instructions can help solve the entity alignment bias issue.
2. How to obtain code-like query instructions.
3. At line 164, why is it said that code-like instructions will not introduce noise?
4. Line 40 mentions "model generalization", however, why introduce model generalization? I can't understand the relationship between model generalization and the proposed method.
5. At line 405, the authors mention that using constraints will accelerate convergence, but the corresponding experimental verification is lacking.
6. At line 441, why are the ranges of these three settings like this?
7. At line 472, the word "reasoning" is confusing. What I understand is that the author wants to express the model prediction stage.
8. In line 1209, "a' spefific'"? (It should be noted that "spefific" here is likely a misspelling, and the correct one might be "specific".)

**Questions:**

Please refer to the above weaknesses part.

**Reviewer Confidence:**

3: The reviewer is confident but not certain that the evaluation is correct

**Scope:**

4: The work is relevant to the Web and to the track, and is of broad interest to the community

---

### Official Review · Reviewer_XSSt · 2024-12-03

**Novelty:** 4
**Technical Quality:** 5

**Review:**

This paper presents an approach to first order logic query answering over incomplete knowledge bases, that uses a structural representation of common query patterns to enhance the query representation of natural language. A plug and play approach is taken where a natural language query is parsed into code like instructions and query are answered through knowledge graph embeddings.

The work follows a large number of recent works addressing logical query answering through knowledge graph embeddings, which have have shown some good performance with basic queries, but tend to struggle as complexity increases, and particularly in cases where negations are used. In this case query patterns corresponding to common forms of knowledge graph query are prebuilt and injected into the knowledge graph embedding.

The architecture is clearly described and some justification is given, along with some mathematical results. The experiments show some incremental improvements to the best performing models, in line with the extra information provided to the model. Some visualisations motivate the improvide a motivation for the improved performance, however the paper lacks a qualitative analysis actual queries to allow the reader to get a sense of the usefulness of the approach.

The paper is generally presented very well and is largely free of errors.
Some concepts need better explanation, it's not clear what compressive optimisation boundaries are or whether they are a good thing.
Theorem 3.1 is poorly phrased, and reads more as a discussion than a statement. "More compact" in continuous space is not a clearly defined, and there's not a clear motivation for why it is a good thing (halving a range can be easily achieved with linear transformations).

This is an interesting and generally well motivated contribution, but only gives incremental improvements and the theoretical contributions are not convincing.

**Questions:**

Can you comment on the robustness of the pattern injection. What happens when queries not conforming to common patterns are posed, and what happens when patterns are misidentified?

**Reviewer Confidence:**

2: The reviewer is willing to defend the evaluation, but it is likely that the reviewer did not understand parts of the paper

**Scope:**

3: The work is somewhat relevant to the Web and to the track, and is of narrow interest to a sub-community